# Transcriptome-wide analysis of the function of Ded1 in translation preinitiation complex assembly in a reconstituted in vitro system

**Fujun Zhou[1], Julie M Bocetti[1†], Meizhen Hou[1†], Daoming Qin[1†‡], Alan G Hinnebusch[2*], Jon R Lorsch[1*]**

[1]Section on the Mechanism and Regulation of Protein Synthesis, Eunice Kennedy Shriver National Institute of Child Health and Human Development, Bethesda, United States; [2]Section on Nutrient Control of Gene Expression, Eunice Kennedy Shriver National Institute of Child Health and Human Development, Bethesda, United States

**\*For correspondence:**
alanh@mail.nih.gov (AGH);
jon.lorsch@nih.gov (JRL)

†These authors contributed equally to this work

**Present address:** ‡Steamify LLC, Mclean, United States

**Abstract** We have developed a deep sequencing-based approach, Rec-Seq, that allows simultaneous monitoring of ribosomal 48S preinitiation complex (PIC) formation on every mRNA in the translatome in an in vitro reconstituted system. Rec-Seq isolates key early steps in translation initiation in the absence of all other cellular components and processes. Using this approach, we show that the DEAD-box ATPase Ded1 promotes 48S PIC formation on the start codons of >1000 native mRNAs, most of which have long, structured 5′-untranslated regions (5′UTRs). Remarkably, initiation measured in Rec-Seq was enhanced by Ded1 for most mRNAs previously shown to be highly Ded1-dependent by ribosome profiling of *ded1* mutants in vivo, demonstrating that the core translation functions of the factor are recapitulated in the purified system. Our data do not support a model in which Ded1acts by reducing initiation at alternative start codons in 5′UTRs and instead indicate it functions by directly promoting mRNA recruitment to the 43S PIC and scanning to locate the main start codon. We also provide evidence that eIF4A, another essential DEAD-box initiation factor, is required for efficient PIC assembly on almost all mRNAs, regardless of their structural complexity, in contrast to the preferential stimulation by Ded1 of initiation on mRNAs with long, structured 5′UTRs.

## eLife assessment

This is an **important** article as it is the first to use a reconstituted translation system to study competition among mRNAs for the initiation machinery. Understanding the principles of the biochemistry of mRNA competition for initiation factors cannot be achieved without such a system. The authors provide **compelling** evidence that Ded1 is required for efficient initiation of highly structured mRNAs. The findings are significant and validate the in vitro reconstituted system by recapitulating the effects of in vivo perturbations of translation initiation by Ded1 mutants.

## Introduction

Translation initiation in eukaryotes begins with the assembly of the 43S preinitiation complex (PIC), containing the small (40S) ribosomal subunit bound to the eIF2·GTP·Met-tRNA$_i$ ternary complex and several initiation factors (eIFs), including eIF1, eIF1A, eIF5, eIF4B, and eIF3. The 43S PIC binds the 5′ end of m⁷G-capped mRNAs with the assistance of the cap-binding protein complex eIF4F, comprised of cap-binding subunit eIF4E, scaffolding and mRNA-binding subunit eIF4G, and DEAD-box RNA

helicase eIF4A. The resulting 43S·mRNA complex, or 48S PIC, scans the 5′-untranslated region (5′UTR) for a suitable start codon, whose selection triggers release of most of the initiation factors. Subsequent joining of the large (60S) ribosomal subunit to the 40S·Met-tRNA$_i$ complex, with the aid of eIF1A and the GTPase factor eIF5B, forms the 80S initiation complex, poised to begin the elongation phase of protein synthesis (*Hinnebusch, 2014*).

The efficiency of translation initiation on specific mRNAs is strongly influenced by the secondary structures in the 5′UTR and the length of the 5′UTR (*Sonenberg and Hinnebusch, 2009*; *Niederer et al., 2022*). It is believed that the inhibitory structures in mRNA 5′UTRs are resolved by DEAD-/H-box RNA helicases (*Hinnebusch, 2014*). However, we and others have demonstrated that eIF4A is required for the efficient recruitment of a variety of mRNAs tested in vitro regardless of their degrees of structure (*Pestova and Kolupaeva, 2002*; *Yourik et al., 2017*). Consistent with this, profiling of translating 80S ribosomes in vivo (Ribo-Seq analysis) in an eIF4A temperature-sensitive yeast mutant (*tif1-ts*) revealed that eIF4A inactivation reduced the relative translation efficiencies of less than 40 mRNAs, despite a strong reduction in bulk polysome assembly (*Sen et al., 2015*), thus suggesting that the majority of mRNAs have similarly strong requirements for eIF4A in yeast cells. Mammalian eIF4A has been suggested to remodel the 40S subunit to enhance PIC attachment to mRNAs and function beyond its role in RNA unwinding (*Sokabe and Fraser, 2017*) and might facilitate the threading of the 5′ end of mRNA into the 40S entry channel (*Kumar et al., 2016*).

In contrast to eIF4A, inactivation of yeast DEAD-box helicase Ded1 in a cold-sensitive mutant (*ded1-cs*) was found to reduce the relative TEs of more than 1100 mRNAs in Ribo-Seq experiments (*Sen et al., 2015*; *Sen et al., 2019*). These Ded1-hyperdependent mRNAs displayed a marked tendency for long and structured 5′UTRs (*Sen et al., 2015*). Profiling of small (40S) ribosomal subunits in the *ded1-cs* mutant provided evidence that Ded1 stimulates the translation of Ded1-hyperdependent mRNAs by promoting attachment of 43S PICs to the mRNA 5′ ends or subsequent scanning to the start codon (*Sen et al., 2019*). Furthermore, in a purified yeast translation initiation system several mRNAs identified as being hyperdependent on Ded1 in vivo also displayed greater stimulation by Ded1 of 48S PIC assembly than observed for either Ded1-hypodependent mRNAs or variants of the hyperdependent mRNAs lacking their 5′UTR secondary structures (*Gupta et al., 2018*). Together, these findings suggest that Ded1 is important particularly for stimulating translation of mRNAs that have long, structured 5′UTRs. Several lines of evidence indicate that Ded1 activity on such mRNAs is enhanced by interaction of its N-terminal domain with eIF4A and eIF4E, and of its C-terminal domain with eIF4G (*Gao et al., 2016*; *Gupta et al., 2018*; *Gulay et al., 2020*), indicating functional coupling between eIF4F and Ded1 in 48S PIC assembly. Other evidence suggests that Ded1 broadly promotes scanning through 5′UTRs by blocking initiation at alternative start codons positioned just upstream of secondary structures that are unwound by Ded1 in vivo, thus allowing the scanning PIC to continue downstream to the main start codon of the mRNA (*Guenther et al., 2018*).

It appears that Ded1 can regulate translation of mRNAs in vivo in ways ostensibly distinct from its functions in promoting PIC attachment or scanning on structured mRNAs. Ded1 and its mammalian ortholog DDX3 have been implicated in facilitating the joining of the large ribosomal subunit to the 48S PIC after start codon recognition (*Geissler et al., 2012*; *Wang et al., 2022*). There is also evidence that Ded1 can repress translation by segregating mRNAs into phase-separated RNA–protein granules, processing bodies (PBs), or stress granules (SGs), during cell stresses. Granule formation in glucose-starved cells is reduced by depletion of Ded1 and is enhanced by Ded1 overexpression or impairment of its ATPase activity in non-starved cells in a manner regulated by regions in the Ded1 NTD and CTD (*Beckham et al., 2008*; *Hilliker et al., 2011*; *Hondele et al., 2019*). It was suggested that Ded1 initially forms an inactive complex with eIF4F and mRNA that is stalled at a step upstream of 43S PIC joining and subsequently utilizes ATP hydrolysis to allow progression into the initiation pathway (*Beckham et al., 2008*; *Hilliker et al., 2011*).

Heat-shock also evokes Ded1 sequestration in SGs in a manner antagonized at low temperatures by a polar intrinsically disordered region (IDR) in the Ded1 NTD. Purified Ded1 displayed heat-induced phase separation in vitro that was enhanced by mRNA and the Ded1 CTD but suppressed by the N-terminal IDR. Interestingly, heat-shock preferentially reduced the TEs of mRNAs with structured 5′UTRs and increased their condensation, in a manner modulated by the N-terminal IDR, leading to the model that condensation of Ded1 and associated mRNAs preferentially reduces the translation of Ded1-hyperdependent mRNAs with structured 5′UTRs during heat-shock (*Iserman et al., 2020*).

Other findings indicate that glucose starvation or heat-shock leads to widespread dissociation of Ded1, eIF4A, and eIF4B from the 5′UTRs of mRNAs (*Castelli et al., 2011*; *Bresson et al., 2020*), which could underlie reduced translation of Ded1-hyperdependent mRNAs that remain soluble in stressed cells. Examining the TE changes conferred by glucose starvation or heat-shock by ribosome profiling suggested that Ded1-hyperdependent mRNAs vary greatly in their translational suppression during stress, which might involve a combinatorial effect of impairing eIF4A, eIF4B, and Ded1 function on the most affected subsets (*Sen et al., 2021*).

It is possible that Ded1 has additional functions in the nucleus as it appears to shuttle between nucleus and cytoplasm and interacts both physically and functionally with the nuclear cap-binding proteins (*Senissar et al., 2014*). Ded1 was also found to co-purify with pre-90S and pre-40S immature ribosomes (*Schäfer et al., 2003*), and thus might function in ribosome biogenesis.

A difficulty with interpreting the previous ribosome profiling experiments with *ded1* mutations is that the inactivated mutant protein could have had dominant effects rather than simply producing a loss of Ded1 function. For example, in view of evidence that inhibition of Ded1 ATPase activity can lead to the formation of translationally inert mRNP granules, it is possible that the *ded1* mutations in the catalytic domain analyzed by ribosome profiling led to impaired translation of many Ded1-hyperdependent mRNAs in non-stressed cells owing to their sequestration with the mutant *ded1* proteins in condensates inaccessible to the initiation machinery. It was also possible that the inactivated mutant proteins bound to mRNAs, the ribosome, or other initiation factors in a manner that inhibited translation. The *ded1* mutations could also indirectly impair translation of Ded1-hyperdependent mRNAs by reducing their export from the nucleus, reducing expression of another protein with a role in production or function of the translation initiation machinery, activating stress responses that alter the functions of initiation factors, or eliciting quality control responses that alter the rate of translation elongation. Even wild-type Ded1 could influence translation indirectly by modulating the association of various RNA-binding proteins with mRNAs, a function ascribed to other DEAD-box proteins (*Linder and Jankowsky, 2011*). As noted above, we demonstrated that several mRNAs that were found to be Ded1-hyperdependent in vivo displayed a greater stimulation of 48S PIC assembly by Ded1 compared to several in vivo hypodependent mRNAs in a fully purified system, confirming a direct role of Ded1 in stimulating initiation on these few mRNAs in the soluble phase in the absence of any other cellular proteins besides ribosomes and canonical initiation factors. However, it was possible that many other mRNAs found to be hyperdependent on Ded1 in vivo are not stimulated directly by Ded1's ability to resolve secondary structures in 5′UTRs and promote PIC recruitment or scanning to the start codon.

We have developed an elaboration of the reconstituted yeast translation initiation system (*Acker et al., 2007*) that enables examination of 48S PIC assembly on all native yeast mRNAs simultaneously. This approach, dubbed Recruitment-Sequencing (Rec-Seq), allowed us to measure direct effects of Ded1 on PIC attachment to mRNA and scanning to the start codon in reactions containing only 40S ribosomal subunits, methionyl-initiator tRNA (Met-tRNA$_i$), GDPNP and ATP, and the core initiation factors. In previous work using the reconstituted yeast initiation system, we found that stable 48S PIC formation on mRNA start codons was strongly dependent on the 5′-cap (*Mitchell et al., 2010*; *Yourik et al., 2017*). The cap also prevented off-pathway events and enforced a requirement for the full set of initiation factors to efficiently assemble a 48S PIC. These results indicate that in the reconstituted system 43S PICs are loaded onto the 5′ ends of mRNAs in a cap-dependent manner and that they scan from 5′ to 3′ to locate the start codon.

Rec-Seq allowed us to isolate the crucial and highly regulated series of events leading up to 48S PIC assembly from all later steps of initiation and elongation. Processes such as mRNA localization and decay and RNA and protein compartmentalization, which can complicate in vivo analyses, likewise do not contribute to the outcome of Rec-Seq experiments. Because we could vary Ded1 concentration in reactions or leave it out entirely, Rec-Seq allowed us to directly interrogate the factor's role instead of relying on inferences from the effects of *ded1* mutations. Moreover, we could evaluate whether preventing initiation at alternative start codons in 5′UTRs is an important aspect of Ded1 enhancement of PIC assembly at the canonical start codons on Ded1-stimulated transcripts in vitro.

Our studies indicate that in the Rec-Seq system Ded1 stimulates 48S PIC assembly on ~1000 mRNAs. These Ded1-dependent mRNAs include ~90% of the mRNAs previously found to be hyperdependent on Ded1 in vivo by ribosome profiling of the *ded1-cs* mutant. The fact that addition of Ded1 stimulates 48S PIC assembly on these mRNAs in the reconstituted Rec-Seq system argues against the

involvement of other cellular processes or proteins beyond the core initiation machinery in Ded1 function, and the possibility that the in vivo ribosome profiling results were influenced by dominant negative effects of the *ded1-cs* mutation or by altered expression or activity of the translational machinery in the mutant cells. The mRNAs that are strongly dependent on Ded1 in the Rec-Seq system exhibit a high propensity for long and structured 5′UTRs, providing compelling evidence that Ded1 acts directly to stimulate 48S PIC formation by unwinding 5′UTR structures to facilitate mRNA binding to the PIC and scanning to find the start codon. Our results also indicate that alternative initiation at upstream start codons is too infrequent, and the suppression of these events by addition of Ded1 too small, to account for the stimulatory effects of Ded1 on 48S PIC assembly in the reconstituted system. Finally, we show that eIF4A strongly stimulates the formation of 48S PICs on the vast majority of yeast mRNAs, regardless of their 5′UTR length or degree of secondary structure, in contrast to the specificity we observe for Ded1 in stimulating recruitment of mRNAs with long, structured 5′UTRs. These data are consistent with the model that eIF4A plays a general role in facilitating binding of all mRNAs to the 43S PIC, whereas Ded1 unwinds inhibitory secondary structures to promote PIC binding and scanning on mRNAs with structured 5′UTRs.

## Results

### Rec-Seq: An in vitro approach to measure 48S PIC formation transcriptome-wide

We previously developed a fully reconstituted yeast translation initiation system (*Acker et al., 2007*) that allows measurement of stable recruitment of mRNAs to 43S PICs (*Mitchell et al., 2010*) to form 48S PICs using defined concentrations of purified components for individual mRNAs. In addition to purified small (40S) ribosomal subunits, reactions include initiation factors eIF1, eIF1A, eIF5, the eIF2·GTP·Met-tRNA$_i$ ternary complex (TC, assembled using non-hydrolyzable GDPNP), eIF3, the mRNA-recruitment factors eIF4G·eIF4E complex, eIF4A and eIF4B, and ATP. In the initial version of this system, we monitored recruitment of a single-capped, radiolabeled, in vitro transcribed mRNA in each reaction. Formation of a stable 48S PIC with the anticodon of Met-tRNA$_i$ base paired to the AUG start codon in the mRNA was resolved from free mRNA by native gel electrophoresis and quantified by phosphorimaging of the mRNA. Among other things, this assay has been used to elucidate the molecular roles of eIFs essential for 48S PIC assembly (*Hinnebusch, 2014*); provide evidence that eIF4A is essential for recruitment of various mRNAs to the 43S PIC regardless of the amount of secondary structure in their 5′UTRs (*Yourik et al., 2017*); and reconstitute the ability of Ded1 to overcome 5′UTR secondary structures found in several Ded1-hyperdependent mRNAs (*Gupta et al., 2018*). An important limitation of the assay, however, was that only a single mRNA could be examined at a time, and the effects of competition among the many different mRNAs that exist in a cell could not be assessed. In addition, the use of an electrophoretic mobility-shift assay (EMSA) limits the analysis to very short native transcripts, for example, *RPL41A* mRNA, or the use of synthetic constructs truncated at their 3′ ends in place of longer, full-length native mRNAs (*Gupta et al., 2018*). Such constructs might not recapitulate functionally important interactions between the 5′UTR and other parts of the mRNA.

To circumvent these limitations, we modified the system to allow 48S PIC formation on all native mRNAs in the standard yeast transcriptome to be monitored simultaneously using an approach we call Rec-Seq for 'Recruitment-Sequencing' (*Figure 1A*). In place of a single, radiolabeled mRNA, we used oligo(dT) affinity-purified total mRNA from wild-type yeast cells grown in nutrient-replete conditions. This total polyA(+) mRNA was incubated for various times with pre-assembled 43S PICs, eIF3, eIF4 factors, ATP, and other components (depending on the experiment; *Figure 1—figure supplement 1*). The reactions were then treated with RNase I to digest mRNA regions not protected by the ribosomal complex. For each reaction, 48S PICs were isolated by sedimentation through a sucrose gradient and ribosome-protected fragments (RPFs) were purified by gel electrophoresis and converted to cDNAs by reverse transcription followed by PCR amplification to generate sequencing libraries. This approach allowed us to monitor PIC assembly on all native yeast mRNAs in parallel.

To make results comparable among replicates and across experiments, we used an internal normalization approach in which 48S PICs were assembled on two non-native mRNAs encoding Renilla or firefly luciferase and treated with RNase I. A constant amount of these 'spike-in' 48S PICs were added

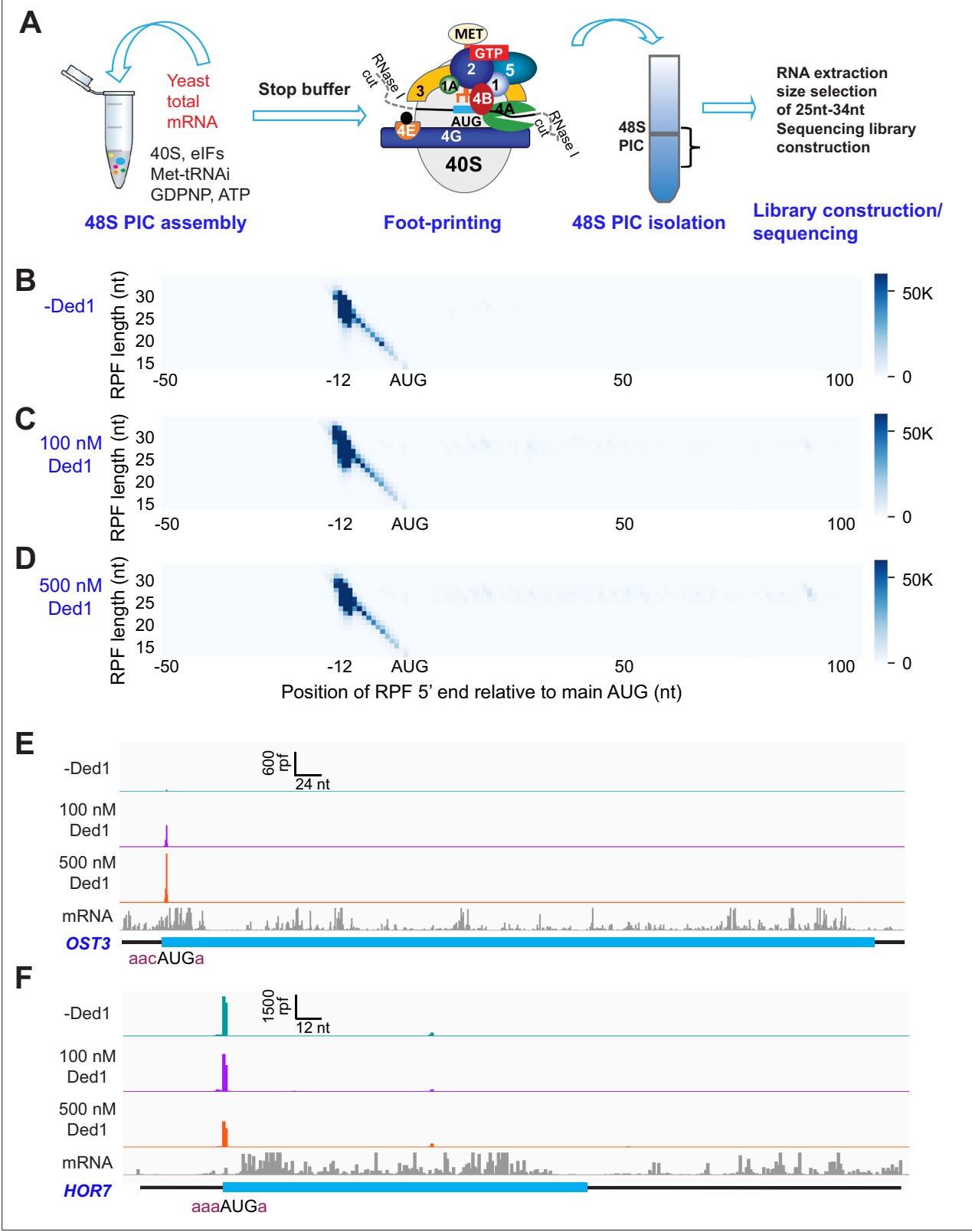

**Figure 1.** Recruitment Sequencing (Rec-Seq) allows transcriptome-wide analysis of early steps of translation initiation in a purified system. (**A**) Overview of steps in the Rec-Seq method. 48S ribosomal preinitiation complexes (PICs) are assembled in vitro from purified *S. cerevisiae* components. 48S PICs are treated with RNase I to digest unprotected mRNA and then isolated using sucrose density gradient ultracentrifugation. Ribosome-protected mRNA fragments (RPFs) are used to construct a sequencing library. (**B–D**) Metagene plots of RPF distributions over 150 nt windows (–50 to +100 nt) on all

*Figure 1 continued on next page*

*Figure 1 continued*

mRNAs aligned to their main AUG start codons for no Ded1 (**B**); 100 nM Ded1 (**C**), or 500 nM Ded1 (**D**). RPF lengths are shown on the Y-axis and the position of the 5′ ends of the RPFs relative to the main start codon are shown on the X-axis. A 5′ end located 12 nt from the start codon is expected for 48S PICs with the AUG in the P site of the 40S subunit (*Wagner et al., 2020*). The color scale shows RPF density. All reads for three replicates for each condition were combined. (**E, F**) RPFs for the 0, 100, and 500 nM Ded1 experiments for previously identified Ded1 hypodependent (*OST3*, **E**) and hyperdependent (*HOR7,* **F**) mRNAs. The position of the main coding sequences (CDS) is shown in cyan and the −3 to −1 and +4 context nucleotides surrounding the main AUG are shown in brick red text. mRNA sequencing reads are also shown below each set of tracks (gray). The Integrated Genome Viewer (IGV, Broad Institute) was used to display RPF and mRNA reads, with RPF and nucleotide (nt) scales indicated on top of each panel. For these and all other gene browser views, the RPFs are plotted to their predicted P-site positions, and the mRNA reads are plotted to their first position from their transcript 5′ ends.

The online version of this article includes the following figure supplement(s) for figure 1:

**Figure supplement 1.** A schematic diagram of mRNA recruitment in translation initiation.

**Figure supplement 2.** Internal spike-in normalization controls using preformed 48S preinitiation complexes (PICs) on non-native mRNAs.

**Figure supplement 3.** Transcriptome-wide reproducibility of ribosome-protected fragments (RPFs) among replicates.

**Figure supplement 4.** Ribosome-protected fragments (RPFs) on main AUGs are consistent among replicates.

to each experimental reaction prior to sucrose gradient sedimentation (*Figure 1—figure supplement 2A*). RPFs corresponding to both the main AUG (mAUG) and several internal AUGs (iAUGs) were found to be highly reproducible among biological replicates for these spike-in controls (*Figure 1—figure supplement 2B–D*). Normalizing results with these control RPFs allowed us to achieve high reproducibility among triplicate biological replicate experiments (*Figure 1—figure supplement 3*) and also allowed us to measure absolute rather than relative 48S PIC formation efficiencies despite differences in final read depths from reaction to reaction.

To assess the function of Ded1 in mRNA recruitment, scanning, and start codon recognition, we carried out Rec-Seq experiments on reactions that contained all of the canonical initiation factors shown in *Figure 1A* (including eIF4A) and either lacked Ded1 or contained 100 or 500 nM Ded1. The reactions were initiated by adding total polyA(+) mRNA to the preformed 43S complexes and other factors, and then quenched rapidly after 15 min by depleting ATP with addition of a 'Stop buffer' containing hexokinase and glucose. We chose 15 min because recruitment has reached its observed endpoint for nearly all mRNAs at this time. The RPFs thus measured give the final distribution of competitive recruitment efficiencies under these experimental conditions, in which there is a twofold excess of total mRNA over 43S PICs. After sequencing cDNA libraries prepared from the RNAse I-treated purified 48S PICs and removing sequencing reads mapping to non-coding RNAs, the reads mapping to protein coding sequences were examined.

A metagene plot of the number of RPF reads versus position of RPF 5′ ends relative to the mAUG codon for each mRNA revealed that the overwhelming majority contain a single peak of RPFs with the bulk of 5′ ends located 12 nt upstream of the mAUG codon (*Figure 1B–D*). This is the position expected for PICs with the 40S P site containing the AUG start codon base paired with the anticodon of Met-tRNA$_i$ (*Ingolia et al., 2009*; *Archer et al., 2016*). A minority of RPFs have 5′ ends closer to the AUG codon (*Figure 1B–D*), which could result from invasion of RNase I into the 40S exit channel in a minor fraction of PICs positioned at the start codon, or from low-level degradation of purified RPFs during library construction. The two reactions containing Ded1 have a somewhat greater abundance of RPFs at the start codons compared to the reaction lacking Ded1 (*Figure 1—figure supplement 4A*).

We examined individual Rec-Seq traces for several mRNAs previously shown through ribosome profiling to be either hyper- or hypodependent on Ded1 for translational efficiency (TE) and that had also been shown to be similarly dependent on Ded1 for 43S PIC recruitment in vitro in the reconstituted yeast initiation system (*Figure 1E and F*, *Figure 1—figure supplement 4*). The Ded1 hyperdependent mRNAs showed strong Ded1-enhancement of Rec-Seq RPF peaks mapping to the mAUG (*Figure 1—figure supplement 4B–E*), as exemplified for *OST3* in *Figure 1E*. In contrast, mRNAs previously shown to be hypodependent on Ded1 by ribosome profiling, including *HOR7* (*Figure 1F*), showed modest stimulation, no enhancement, or even diminished RPF counts at the mAUG in reactions containing 100 or 500 nM Ded1 versus no Ded1 (*Figure 1—figure supplement 4F–I*).

We conducted differential expression analysis using the software DESeq2 (*Love et al., 2014*) to identify mRNAs that exhibit statistically significant changes in normalized RPFs mapping to the main

start codon (mRPFs) on addition of Ded1. Groups of ~1000 transcripts among the 3052 for which Rec-Seq data were obtained showed more than twofold increased mRPFs in the presence of 100 or 500 nM Ded1 compared to no added Ded1, whereas only ~100 transcripts showed higher mRPFs at 500 nM versus 100 nM Ded1 (*Figure 2A–C*, red dots). Substantially smaller numbers of mRNAs displayed decreased mRPFs as Ded1 concentration was increased (*Figure 2A–C*, dark blue dots; 71, 182, and 51 mRNAs, respectively). Importantly, 87% of mRNAs showing significant increases in mRPFs at 100 nM Ded1 were also increased at 500 nM Ded1 (*Figure 2D*), and 68% of the mRNAs with significantly diminished mRPFs at 100 nM Ded1 were also decreased at 500 nM Ded1 (*Figure 2E*)—both highly significant overlaps. These data indicate consistency in the effects of Ded1 observed in different experiments. The 1006 mRNAs with significantly increased mRPFs at 500 nM Ded1 had median increases of 3.6-fold and 4.8-fold at 100 and 500 nM Ded1, respectively (*Figure 2F*, cols. 3–4). Similarly, the 911 mRNAs that had significantly increased mRPFs in 100 nM Ded1 had median increases of 4.1-fold and 5.2-fold at 100 and 500 nM Ded1, respectively, and the 793 mRNAs common to both sets had median increases of 4.7-fold and 6.1-fold (*Figure 2F*, cols. 5–8). Although the greater median stimulation at 500 nM versus 100 nM Ded1 is significant for all three mRNA groups, these results indicate that 100 nM Ded1 is nearly saturating for enhancement of 48S PIC assembly for the majority of Ded1-stimulated mRNAs in the in vitro system.

The set of mRNAs displaying diminished mRPFs at 500 nM Ded1 showed smaller reductions at 100 nM (*Figure 2F*, cols. 9–10; median mRPF changes of 0.59 versus 0.37 at 100 and 500 nM Ded1, respectively), indicating that the inhibitory effect of Ded1 on these mRNAs is not saturated at the lower concentration. For the remaining two groups that showed Ded1 suppression of mRPFs at 100 nM or both 100 and 500 nM Ded1, the median degree of Ded1 inhibition did not differ significantly between the two concentrations (*Figure 2F*, cols. 11–14), indicating that inhibition by Ded1 was saturated at 100 nM.

The mRPF reads for each gene were normalized to the total RNA reads across the coding sequences (CDS) of the mRNA (normalized for CDS length)—a measure of transcript abundance—to calculate the recruitment efficiency (RE) of each transcript. The RE is analogous to TE determined by in vivo Ribo-Seq experiments. We examined changes in RE between experiments containing different concentrations of Ded1 by ordering mRNAs according to the change in RE between 0 and 100 nM Ded1 and then using this ordering to generate a heat map depiction of ΔRE values for all 3052 transcripts observed by Rec-Seq between 0 and 100 nM Ded1, 0 and 500 nM Ded1, and 100 and 500 nM Ded1 (*Figure 2G*). Consistent with *Figure 2D and E*, the results indicate that most mRNAs with increased REs on addition of 100 nM Ded1 also have increased RE at 500 nM Ded1 (compare red lines in cols. 1–2), which for certain mRNAs is of greater magnitude at 500 nM (relatively darker red bars in col. 2). The decrease in RE observed for a smaller number of mRNAs upon Ded1 addition (*Figure 2G*, blue lines) generally appears to be greater at 500 nM versus 100 nM Ded1 (cf. blue lines in cols. 1–2), which is also consistent with the results in *Figure 2E and F*.

Examining the distributions of all mRNAs across 10 bins of increasing RE values at each concentration of Ded1 revealed an obvious shift of mRNAs from bins of lowest RE to bins of higher RE when Ded1 was included in the Rec-Seq reactions (*Figure 2H*, cf. orange versus blue and magenta bars, particularly in the first three bins on the left). The similar distributions observed for the 100 and 500 nM Ded1 data are consistent with results in *Figure 2F and G*, indicating that stimulation by Ded1 is nearly saturated at 100 nM. These data suggest that Ded1 confers relatively greater stimulation for mRNAs that are recruited by 43S PICs poorly in the absence of Ded1. Supporting this conclusion, the increase in RE between reactions with 500 nM versus no Ded1 was the greatest for the subset of mRNAs of lowest RE and the least for the mRNAs with highest RE in reactions without Ded1 (*Figure 2I*). This finding recapitulates and extends previous results on several individual mRNAs wherein Ded1 strongly stimulated 43S PIC recruitment of Ded1 hyperdependent transcripts exhibiting poor recruitment in reactions lacking Ded1, while more weakly stimulating Ded1 hypodependent mRNAs that could be recruited efficiently without Ded1 (*Gupta et al., 2018*).

In summary, the Rec-Seq results are highly consistent among biological replicates and reproducibly identify a subset of ~1000 of the 3052 mRNAs detected in the analysis for which assembly of 48S PICs is strongly stimulated by Ded1, with a marked tendency for greater stimulation for mRNAs that recruit PICs poorly in Ded1's absence. These data support previous conclusions from in vivo ribosome profiling analysis of a *ded1-cs* mutant that a significant number (~600 to ~1100) of yeast mRNAs

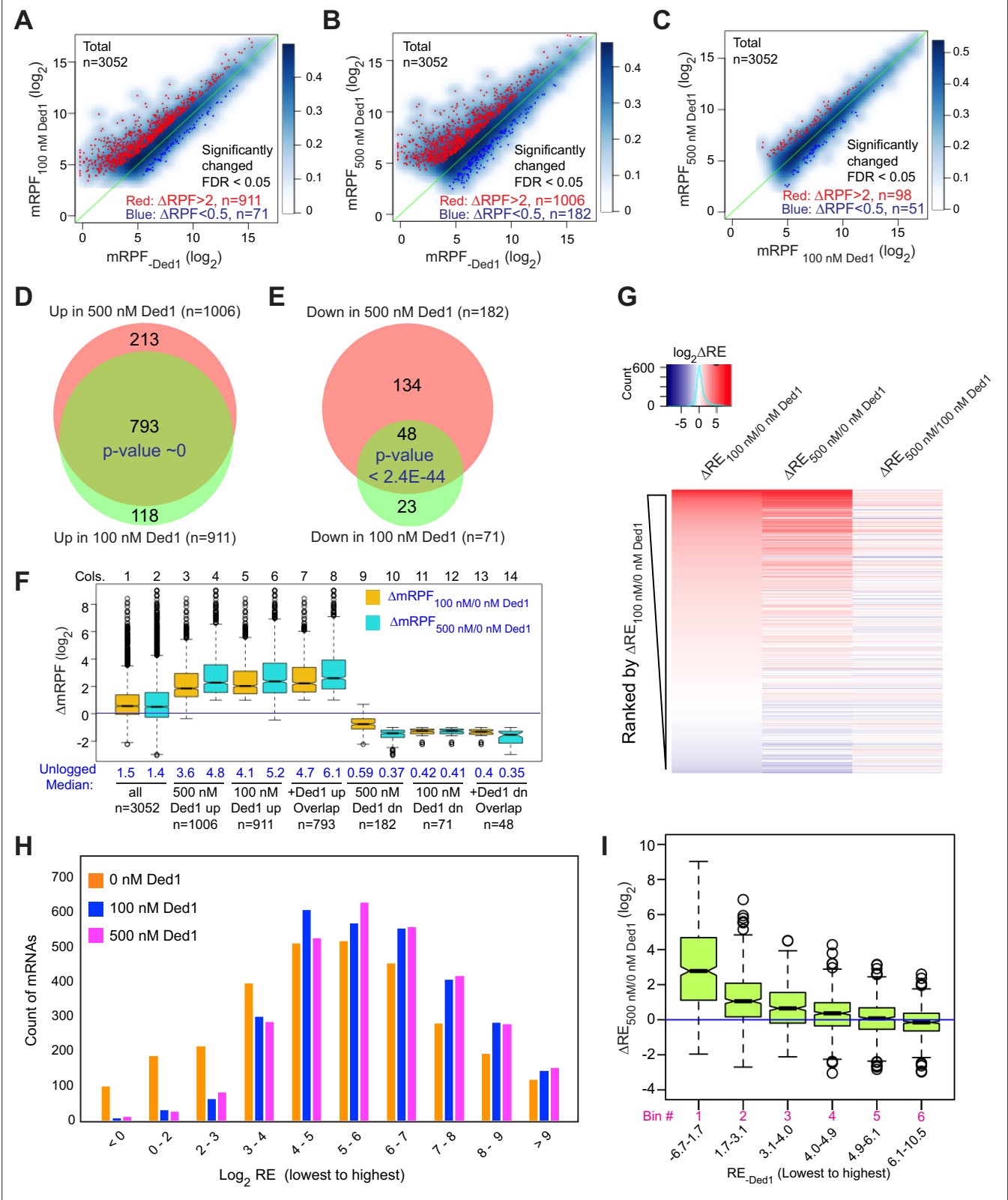

**Figure 2.** Ded1 promotes the recruitment of a group of 1000 mRNAs. (**A–C**) Scatterplots of normalized read densities on the main start codons of mRNAs (mRPF, number of ribosome-protected fragments [RPFs] mapping to the main AUG normalized by the geometric means of the spike-in internal control RPFs) for 3052 mRNAs with >90 total reads in nine samples (three replicates each for 0, 100, and 500 nM Ded1) for 100 versus 0 nM Ded1 (**A**), 500 versus 0 nM Ded1 (**B**), or 500 versus 100 nM Ded1 (**C**). mRNAs with significantly changed recruitment were defined as those with false discovery

*Figure 2 continued on next page*

*Figure 2 continued*

rate (FDR) < 0.05 and RPF changes (ΔRPF) of >2 for increased or <0.5 for decreased are indicated in red or dark blue, respectively. The number of significantly changed mRNAs is indicated in red (increased) and blue (decreased). (**D, E**) Overlaps between the mRNAs displaying significantly increased (**D**) or significantly reduced (**E**) mRPFs between 100 and 500 nM Ded1 conditions. p-Values for the overlap between each group were calculated using an online tool (http://nemates.org/MA/progs/overlap_stats.html). (**F**) Boxplot analysis of the distribution and median of $\log_2$ change in mRPF between 0 and 100 nM Ded1 ($\Delta mRPF_{100/0\ nM\ Ded1}$, orange) or 0 and 500 nM Ded1 ($\Delta mRPF_{500/0\ nM\ Ded1}$, cyan) for all 3052 mRNAs (cols. 1–2), and for the following 6 groups of mRNAs: 1006 or 911 for which mRPFs were significantly up in 500 or 100 nM Ded1, respectively, relative to no Ded1 (cols. 3–6, for red dots in **B** and **A**); 793 mRNAs for which mRPFs were significantly up with both 100 and 500 nM Ded1 (cols. 7–8, for the overlapping mRNAs in **D**); 182 and 71 mRNAs for which mRPFs were significantly down in 500 or 100 nM Ded1 relative to no Ded1 (cols. 9–12, blue dots in **B** and **A**); and 48 mRNAs for which mRPFs were significantly down with both 100 and 500 nM Ded1 (cols.13–14, for the overlaps in **E**). The blue horizontal line shows no change in mRPFs ($\Delta mRPF$ = 1). (**G**) Heat map analysis of the recruitment efficiency changes $\Delta RE_{100/0\ nM\ Ded1}$, $\Delta RE_{500/0\ nM\ Ded1}$ and $\Delta RE_{500/100\ nM\ Ded1}$ for the 3052 mRNAs described in (**A–C**), ordered by rank of $\Delta RE_{100/0\ nM\ Ded1}$ from most increased (top) to most decreased (bottom) using the R heatmap.2 function. (**H**) The distributions of all observed mRNAs across 10 bins of increasing RE values at each concentration of Ded1 (0, 100, 500 nM). (**I**) Boxplot analysis of $\Delta RE_{500/0\ nM\ Ded1}$ for all 3052 observed mRNAs binned by RE in the absence of Ded1 ($RE_{-Ded1}$) from the lowest to the highest.

exhibit heightened dependence on Ded1 relative to the average mRNA (Ded1-hyperdependent transcripts) (*Sen et al., 2015*; *Sen et al., 2019*). The previous ribosome profiling studies also identified transcripts exhibiting increased relative TEs in *ded1* mutant cells deficient in Ded1 activity (Ded1-hypodependent transcripts), consistent with translational repression by wild-type Ded1. It was unclear, however, whether the translational efficiencies of these last mRNAs were truly elevated in *ded1* cells or merely reduced by a smaller amount than the average transcript to yield an increase in relative TE. Owing to the spike-in normalization employed here, it is clear that the absolute occupancies of 48S PICs at mAUGs on mRNAs showing reduced REs on Ded1 addition are indeed diminished by Ded1, although it remains unclear whether the repression is direct or results indirectly from increased competition for limiting PICs conferred by enhanced PIC recruitment on the Ded1-stimulated mRNAs. We examine this question further below.

## Transcripts showing strong Ded1 stimulation of 48S PIC assembly in Rec-Seq have long and structured 5′UTRs

The mRNAs judged to be hyperdependent on Ded1 from Ribo-Seq experiments on *ded1* mutants are enriched for transcripts with longer than average 5′UTRs with a heightened propensity for forming secondary structures (*Sen et al., 2015*). To examine these trends in our in vitro system, we divided the subset of all transcripts analyzed by Rec-Seq with annotated lengths for their predominant 5′UTRs (N = 2804, *Pelechano et al., 2013*) into six bins of equal size and plotted their average REs in the presence or absence of Ded1. In the absence of Ded1, the average RE values for these bins is roughly constant for the first four 5′UTR length bins and then decreases markedly in the fifth and sixth bins (*Figure 3A*, orange). The addition of Ded1 differentially enhanced RE values depending on 5′UTR length, conferring the largest increase for transcripts with the longest 5′UTRs (bin 6), smaller increases for bins 3–5 having the next longest 5′UTRs, and little or no increase for the mRNAs in bins 1–2 with the shortest 5′UTRs (*Figure 3A*, cyan and purple versus orange). The 5′UTR length trends were confirmed by plotting median RE changes conferred by 100 or 500 nM Ded1 versus no Ded1 for all six bins (*Figure 3B and C*), which showed little or no stimulation in bins 1–2 and progressively larger increases in bins 3–6. Consistent with this, comparing 500 nm to 100 nm Ded1 (*Figure 3D*) revealed that the higher concentration of Ded1 actually reduced the median REs of the mRNAs with shortest 5′UTRs in bins 1–2, while increasing it for the transcripts with longest 5′UTRs in bins 5–6.

To examine the correlation between 5′UTR secondary structure and the effects of Ded1 on RE values, we used a compilation of propensities for secondary structure in the yeast transcriptome (*Kertesz et al., 2010*) in which each nucleotide in 3000 different yeast transcripts was assigned a 'parallel analysis of RNA structure' (PARS) score based on its susceptibility to digestion with single- or double-stranded specific nucleases in yeast mRNA reannealed in vitro. In this analysis, a higher PARS score denotes a higher probability of double-stranded conformation. We calculated various cumulative PARS scores for 5′UTRs, including the sum of scores for (i) all 5′UTR nucleotides (total PARS); (ii) the 30 nt surrounding the start codon (Start30 PARS; for mRNAs with a 5′UTR ≥ 15 nt); and (iii) the highest cumulative score in any 30 nt window (Max30 PARS). For each sequence interval, the 1874 transcripts observed in Rec-Seq with available PARS data were divided into six bins of increasing PARS scores and examined for RE changes conferred by Ded1 in Rec-Seq.

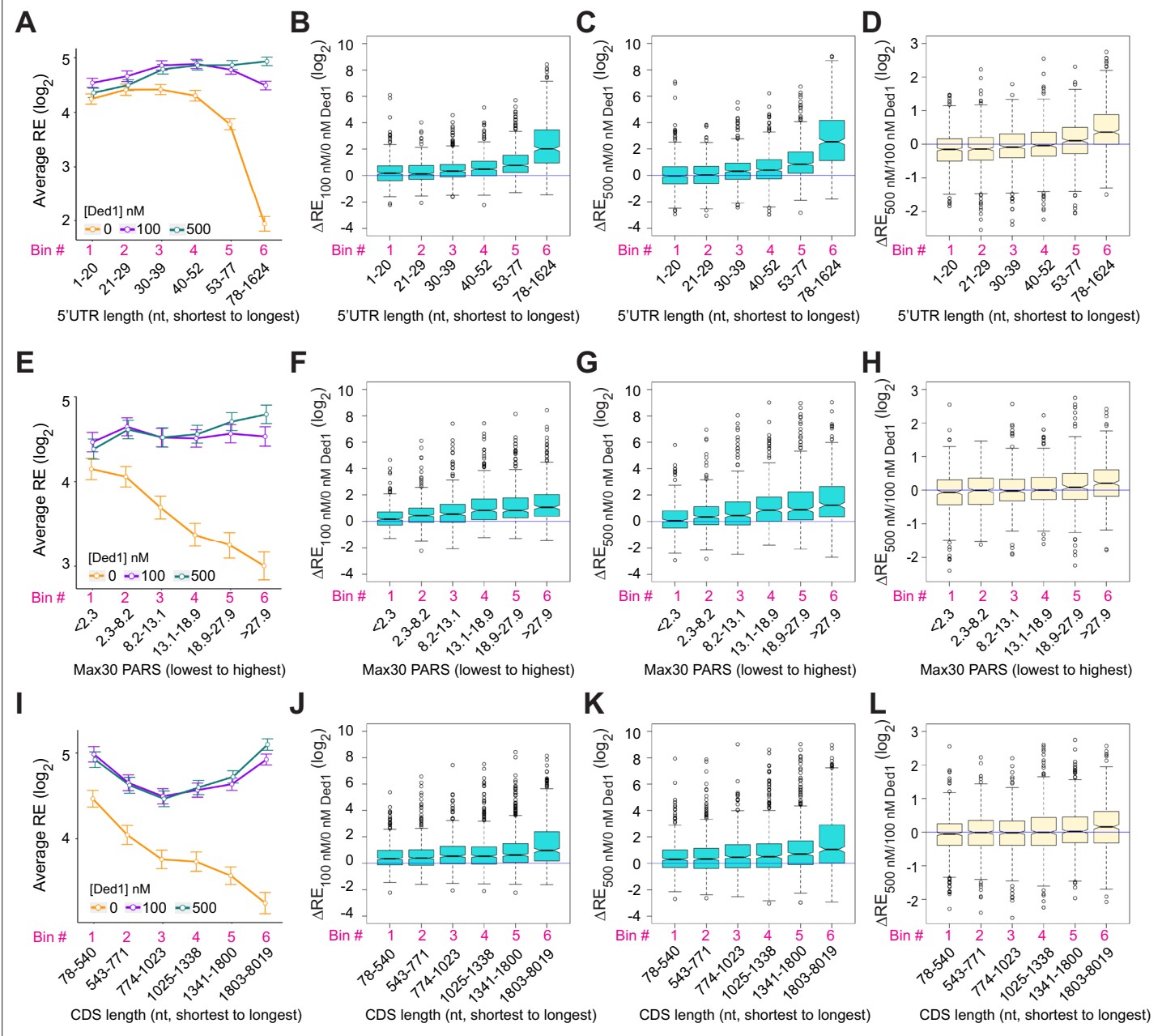

**Figure 3.** Ded1 promotes the recruitment of mRNAs with long or structured 5'UTRs. (**A**) Line plot analysis of average recruitment efficiency (RE) for the 2804 mRNAs observed in the Rec-Seq experiments that have annotated 5'UTRs binned by 5'UTR length at 0, 100, and 500 nM Ded1. Average RE values were determined from the RE values of all mRNAs in each bin. (**B–D**) Boxplot analyses of the RE changes (ΔRE) between each of two different conditions for the same mRNA bins as in (**A**) for $ΔRE_{100/0nM\_Ded1}$ (**B**), $ΔRE_{500/0nM\_Ded1}$ (**C**), or $ΔRE_{500/100nM\_Ded1}$ (**D**). (**E**) Line plot analysis of average RE values for the 1874 mRNAs observed in the Rec-Seq experiments with reported parallel analysis of RNA structure (PARS) scores binned according to Max30 PARS scores from the lowest to the highest. PARS scores were determined by *Kertesz et al., 2010* and Max30 and other PARS score intervals were calculated as described in *Sen et al., 2015*. (**F–H**) Boxplot analyses for the same mRNA bins as in (**E**) for $ΔRE_{100/0nM\_Ded1}$ (**F**), $ΔRE_{500/0nM\_Ded1}$ (**G**), or $ΔRE_{500/100nM\_Ded1}$ (**H**). (**I**) Line plot analysis of average REs for all 3052 mRNAs divided into six equal-sized bins according to coding sequences (CDS) lengths from the shortest to the longest for 0, 100, and 500 nM Ded1. (**J–L**) Boxplot analysis for the same mRNA bins as in (**I**) for $ΔRE_{100/0nM\_Ded1}$ (**J**), $ΔRE_{500/0nM\_Ded1}$ (**K**), or $ΔRE_{500/100nM\_Ded1}$ (**L**). All bins contain an equal number of mRNAs.

The online version of this article includes the following figure supplement(s) for figure 3:

**Figure supplement 1.** Ded1 preferentially stimulates 48S preinitiation complex (PIC) formation on structured mRNAs.

**Figure supplement 2.** Ded1 promotes the recruitment of mRNAs with long 5'UTRs independent of coding sequences (CDS) lengths.

**Figure supplement 3.** Ded1 has little effect on discriminating main AUG context scores.

The average REs of mRNAs binned according to the 5′UTR Max30 PARS scores decline steadily with increasing PARS scores in Rec-Seq reactions lacking Ded1 (*Figure 3E*, orange). Importantly, the negative effect of increasing Max30 PARS scores was essentially eliminated by addition of 100 or 500 nM Ded1 (*Figure 3E*, purple and cyan). Consistent with this, addition of 100 or 500 nM Ded1 confers significant increases in median RE that get progressively larger with increasing Max30 PARS scores (*Figure 3F and G*). Increasing Ded1 from 100 to 500 nM modestly increased the median RE for mRNAs in the two highest Max30 PARS score bins (*Figure 3H*). Similar effects on average RE values were observed for 5′UTR total PARS and Start30 PARS scores (*Figure 3—figure supplement 1A–D*), with relatively greater Ded1 stimulation for the mRNAs with larger scores for both parameters (*Figure 3—figure supplement 1C and D*). CDS length also correlated with diminished average RE values (*Figure 3I*), and this effect was reversed by Ded1 for the mRNAs with longest CDS lengths in the last two or three bins at both 100 and 500 nM Ded1 (*Figure 3I–L*). We do not know why the average RE versus CDS length curves are U-shaped in the presence of Ded1 (*Figure 3I*, green and purple lines). The correlation between Ded1 stimulation of RE and CDS length could be indirect because CDS length also correlates with 5′UTR length such that mRNAs with longer CDSs also tend to have longer 5′UTRs (*Figure 3—figure supplement 1E and F*). Importantly, correlations between Ded1 stimulation and 5′UTR lengths are evident for all three groups of mRNAs containing distinct ranges of CDS lengths (*Figure 3—figure supplement 2A–C*). In contrast, a marked correlation between Ded1 stimulation and CDS length was detected only for the group of mRNAs with longest 5′UTRs (*Figure 3—figure supplement 2D–F*), and only the latter group showed a clear correlation between 5′UTR length and CDS length (*Figure 3—figure supplement 2G–I*). Thus, the correlation between Ded1 stimulation and CDS length appears to be indirect, driven by the tendency for the mRNAs with the longest 5′UTRs to also have correspondingly longer CDSs. Overall, the Rec-Seq results suggest that Ded1 efficiently overcomes the impediment to 48S PIC assembly posed by structures within 5′UTRs, which are cumulative in longer 5′UTRs. This agrees with the conclusion reached from Ribo-Seq analysis of *ded1* mutants in which Ded1-hyperdependent mRNAs were found to have significantly higher median PARS scores for all 5′UTR intervals tested as well as longer than average 5′UTR lengths (*Sen et al., 2015*).

We next examined the dependence of Ded1 enhancement of 48S PIC assembly on the sequences surrounding the start codon. The context scores of AUGs, $AUG_{CAI}$, quantify the similarity between the –6 to +3 positions surrounding a given AUG to the start codons of the 2% most highly translated yeast mRNAs (*Zur and Tuller, 2013*). These context scores range from ~0.16 (poorest) to ~0.97 (best) among all yeast mAUG codons. Binning mRNAs by context score reveals a steady increase in RE with increasing context scores in Rec-Seq reactions lacking Ded1 (*Figure 3—figure supplement 3A*, orange), indicating that good sequence context around the AUG codon promotes 48S PIC formation. However, inclusion of 100 or 500 nM Ded1 increases the average and median RE values similarly for all bins of context scores (*Figure 3—figure supplement 3A*, cyan and purple; *Figure 3—figure supplement 3B and C*). We conclude that the stimulatory effects of Ded1 on 48S PIC formation are independent of the sequence context surrounding the AUG codon. Thus, Ded1 preferentially stimulates recruitment of mRNAs burdened with structured 5′UTRs but not with poor AUG sequence context.

## Transcripts showing Ded1 stimulation of 48S PIC assembly in Rec-Seq include the majority of Ded1-hyperdependent mRNAs identified by Ribo-Seq analysis of a *ded1* mutant

We next asked whether mRNAs exhibiting stimulation of 48S PIC assembly by Ded1 in Rec-Seq include those judged to be hyperdependent on Ded1 in vivo by Ribo-Seq analysis of the *ded1-cs* mutant (*Sen et al., 2015*). First, to assess the similarity between the sets of mRNAs examined in our Rec-Seq experiments and in previous Ribo-Seq experiments, we plotted RNA-Seq reads for all mRNAs observed in Rec-Seq versus those from WT or *ded1-cs* yeast strains in the Ribo-Seq experiments and found that they were very strongly linearly correlated with Spearman coefficients ($\rho$) of 0.90 and 0.88, respectively (*Figure 4A and B*). The mRNAs from the WT and *ded1-cs* strains identified in Ribo-Seq experiments were also well correlated ($\rho$ = 0.97) (*Figure 4C*). Thus, despite having been prepared using somewhat different methods, the total mRNA used in our Rec-Seq experiments was very similar in sequence abundance to that observed in previous in vivo ribosome profiling experiments.

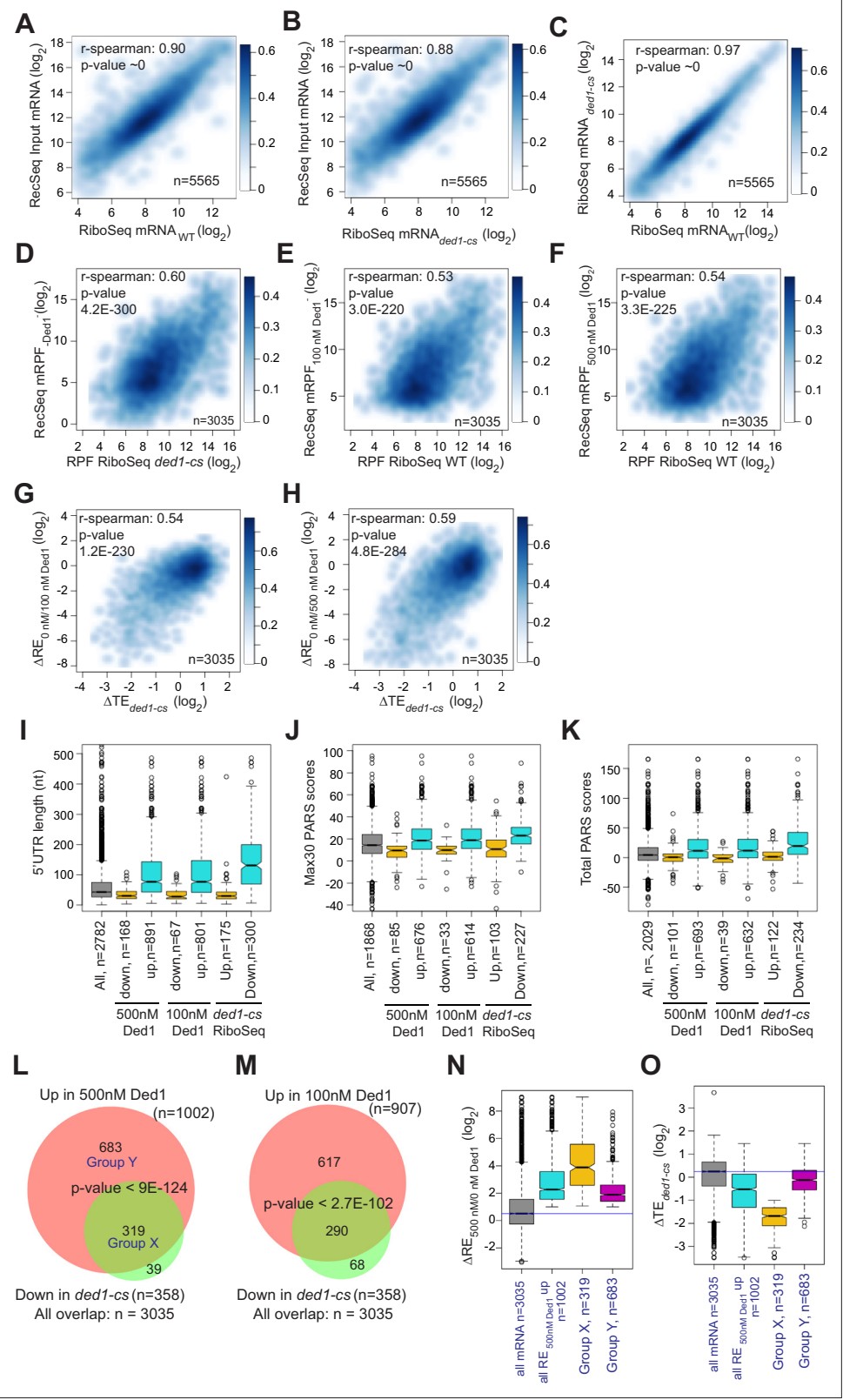

**Figure 4.** Ded1 dependencies observed in Rec-Seq experiments correlate strongly with in vivo results from previous ribosome profiling of the *ded1-cs* mutant. (**A, B**) Scatterplots comparing Rec-Seq input RNA reads and ribosome profiling mRNA reads of WT (**A**) or *ded1-cs* (**B**) strains. (**C**) Scatterplot comparing ribosome profiling mRNA reads between WT and *ded1-cs* strains. (**D–F**) Scatterplots comparing 48S preinitiation complex (PIC)

*Figure 4 continued on next page*

*Figure 4 continued*

ribosome-protected fragments (RPFs) from Rec-Seq to 80S RPFs from ribosome profiling for the 3035 mRNAs that passed the significance cutoffs in both experiments for Rec-Seq at 0 nM Ded1 versus ribosome profiling of the *ded1-cs* mutant (**D**) or Rec-Seq at 100 nM (**E**) or 500 nM (**F**) Ded1 versus ribosome profiling of the WT strain. (**G, H**) Scatterplots comparing changes in translational efficiency (ΔTE) from ribosome profiling of the *ded1-cs* mutant versus WT to $\Delta RE_{0/100nM\_Ded1}$ (**G**) or $\Delta RE_{0/500nM\,Ded1}$ (**H**) values from Rec-Seq. (**I–K**) Boxplot analysis of 5′UTR lengths (**I**), Max30 parallel analysis of RNA structure (PARS) (**J**), or total PARS (**K**) for mRNAs with annotated 5′UTR lengths or PARS scores that showed significantly decreased (Down) or increased (Up) REs in 100 or 500 nM Ded1 versus no Ded1, or significantly increased (Up) or decreased (Down) TEs in the *ded1-cs* mutant versus WT. (**L, M**) Overlaps between mRNAs identified by Rec-Seq at 100 nM Ded1 (**L**) or 500 nM Ded1 (**M**) with significantly increased recruitment efficiencies (REs) versus no Ded1 in Rec-Seq and mRNAs with significantly increased translational efficiencies (TEs) in ribosome profiling of the *ded1-cs* mutant versus WT. p-Values for the overlaps were calculated as described in *Figure 2D and E*. Group X and Y mRNAs in (**L**) are those in common between the two groups being compared (X) or those found exclusively in the set of 1002 mRNAs with significantly increased RE in 500 nM Ded1 versus no Ded1 in Rec-Seq (Y). (**N, O**) Boxplot analysis of $\log_2 \Delta RE_{500/0nM\_Ded1}$ (**N**) or $\log_2 \Delta TE_{ded1-cs}$ (**O**) values for all 3035 mRNAs observed in both Rec-Seq and ribosome profiling, the 1002 mRNAs with significantly increased recruitment with 500 nM Ded1 versus no Ded1 (of **L**), the 319 mRNAs in group X (of **L**), and the 683 mRNAs in group Y (of **L**).

Importantly, we observed a highly significant correlation ($\rho$ = 0.60, p=4.2 × 10⁻³⁰⁰) between the mRPFs from Rec-Seq reactions lacking Ded1 and 80S RPFs from Ribo-Seq experiments with the *ded1-cs* mutant at the non-permissive temperature where Ded1 function is impaired (*Figure 4D*). We saw similar correlations between mRPFs from Rec-Seq reactions containing 100 or 500 nM Ded1 and 80S RPFs from Ribo-Seq experiments on the WT *DED1* strain (*Figure 4E and F*; $\rho$ = 0.53 and $\rho$ = 0.54, respectively). A total of 3035 mRNAs were detected in common across all these experiments. Thus, a marked correlation exists between the amounts of translation on each mRNA observed in vivo (80S RPFs) measured in Ribo-Seq experiments and 48S PIC formation measured in our in vitro Rec-Seq experiments, and this holds at different levels of Ded1 activity.

We also observed significant correlations between ΔRE values measured in Rec-Seq for reactions containing no Ded1 versus 100 or 500 nM Ded1 and changes in translational efficiencies (ΔTE values) measured in Ribo-Seq for *ded1-cs* versus WT *DED1* cells at the non-permissive temperature (*Figure 4G and H*). Consistent with this, the mRNAs exhibiting increased REs on addition of either concentration of Ded1 and the mRNAs showing decreased TEs in *ded1-cs* versus *DED1* cells in Ribo-Seq all have significantly higher median 5′UTR lengths, Max30 PARS scores, or total PARS scores compared to all observed mRNAs and to the mRNAs that behave oppositely in Rec-Seq or Ribo-Seq (*Figure 4I–K*). Strikingly, 80–90% of the mRNAs showing significantly decreased TEs conferred by *ded1-cs* in Ribo-Seq displayed significantly increased REs on addition of 100 or 500 nM Ded1 in Rec-Seq reactions (*Figure 4L and M*).

Despite the significant correlations shown in *Figure 4A–H*, it should be noted that 11% of the Ded1-hyperdependent mRNAs showing TE reductions in *ded1-cs* cells in Ribo-Seq did not exhibit significant stimulation of 48S PIC assembly by Ded1 in our Rec-Seq experiments (*Figure 4L*, set of 39 mRNAs). Such mRNAs might be stimulated by Ded1 at a step following 48S PIC assembly that is not monitored by Rec-Seq or they might require additional factors for Ded1 stimulation of 48S PIC assembly that are lacking in our Rec-Seq experiments. It is also evident that the majority of mRNAs stimulated by 500 nM Ded1 in our Rec-Seq experiments were not judged to be Ded1-hyperdependent in Ribo-Seq analysis of the *ded1-cs* mutant (*Figure 4L*, set of 683 mRNAs; 'Group Y'). This could be explained by the fact that TEs in the Ribo-Seq analysis of the *ded1-cs* mutant were measured relative to the average transcript (*Sen et al., 2015*) and thus both Ded1 hyper- and hypodependent transcripts could have decreased absolute translational efficiencies in the Ded1-deficient mutant at the non-permissive temperature relative to WT cells, but with the former decreased more than the average transcript and the latter decreased similar to or less than the average. Consistent with this interpretation, bulk polysome assembly is dramatically reduced only minutes after shifting the *ded1-cs* mutant to the non-permissive temperature, indicating a reduction in translation initiation on a large fraction of mRNAs (*Sen et al., 2015*). Because the Rec-Seq method employs an internal standard, the measured REs are absolute rather than relative and thus lead to a larger set of mRNAs with increased RE upon Ded1 addition than that for the relative TE reductions in the *ded1-cs* mutant

at the non-permissive temperature in the Ribo-Seq experiments. Consistent with this explanation, the 683 Group Y mRNAs that were significantly stimulated in Rec-Seq by 500 nM Ded1 but not classified as Ded1-hyperdependent in Ribo-Seq displayed a smaller, albeit still significant, median increase in RE upon addition of 500 nM Ded1 than did the 319 mRNAs ('Group X') that were classified as Ded1 hyperdependent in the Ribo-Seq experiments (*Figure 4N*). Moreover, the Group Y mRNAs had an approximately threefold smaller median decrease in relative TE between the *ded1-cs* mutant and WT cells than did the Ded1 hyperdependent Group X mRNAs (*Figure 4O*). The median TE change of Group Y mRNAs was less than twofold and thus would not have met the criteria used for significance in the Ribo-Seq experiments. Thus, normalization of TE changes to the average mRNA in Ribo-Seq undoubtedly obscured the Ded1-dependent in vivo of many of the mRNAs stimulated by Ded1 in Rec-Seq reactions.

Together, our findings suggest that despite the much greater complexity of the system in vivo in terms of both components and processes, the effects of Ded1 on the translation of many mRNAs in cells are similar to what we see in the reconstituted Rec-Seq system that contains only a core set of translation components and is not influenced by processes such as mRNA transport, decay, and compartmentalization/phase separation. This seems a remarkable result given the number of in vivo roles that have been ascribed to Ded1 beyond simply promoting mRNA recruitment and scanning. The significant correlation between the effects of adding Ded1 to Rec-Seq experiments in vitro and of inactivating Ded1 in Ribo-Seq in vivo leads to the important insight that 48S PIC formation is frequently the rate-limiting step for translation initiation that is stimulated by Ded1 in vivo.

## Evidence that Ded1 can promote leaky scanning of canonical AUG start codons

In Rec-Seq reactions lacking Ded1, RPFs at internal AUG start codons (iAUGs) located within CDSs (iRPFs) occur at 3.0% of all RPFs found anywhere in the CDS (at either the main or internal start sites, cdsRPFs), but this proportion increases to 4.2 and 7.8% on addition of 100 or 500 nM Ded1 to the reactions (*Figure 5A*). The increase in iRPFs as Ded1 concentration increases is shown clearly by meta-gene plots for all transcripts (*Figure 5B–D*) and for four specific genes in *Figure 5E–H*, three of which also illustrate that Ded1 can increase iRPFs at genes where 48S PIC assembly at the mAUG is not stimulated or is even reduced by Ded1 (*Figure 5E–G*). DESeq2 analysis reveals that Ded1 elicits more than twofold increases in iRPFs for 590 and 1586 mRNAs at 100 or 500 nM Ded1, respectively, whereas only one or two transcripts showed reduced iRPFs on addition of Ded1, out of all 3052 transcripts detected (*Figure 5I and J*). Thus, induction of iRPFs by Ded1 is widespread in Rec-Seq.

A hallmark of increased leaky scanning in an mRNA would be a reciprocal decrease in mRPF when an increase in iRPF occurs (*Figure 5—figure supplement 1A*). This situation, particularly well-illustrated for *ETT1* and *DBP5* in *Figure 5E and F*, is consistent with Ded1-stimulated readthrough of the mAUG codon, leading to PICs continuing to scan downstream and initiating at iAUGs instead. Importantly, a reciprocal reduction in mRPFs and increase in iRPFs of comparable magnitude on Ded1 addition is evident for the group of 182 mRNAs identified above (*Figure 2B*) showing more than twofold reduction of mRPFs at 500 nM Ded1 (*Figure 5K*). Moreover, 125 (69%) of these 182 transcripts belong to the group of 1586 mRNAs identified in *Figure 5J* displaying more than twofold increases in iRPFs at 500 nM Ded1 (*Figure 5L*, green/red intersection). These 125 transcripts are shown as black dots in the scatterplot of changes in mRPFs at 500 nM versus 0 nM Ded1 in *Figure 5M*. To apply a stringent criteria for leaky scanning, we identified the mRNAs among this last group of 125 mRNAs for which the increase in iRPFs is comparable in magnitude (>50%) to the decrease in mRPFs in the manner expected if iRPFs arise from PICs that scan past the mAUG and initiate internally instead (highlighted with yellow circles in *Figure 5M*). An important outcome of this analysis is that leaky scanning of the mAUG could account for at least 46% (84/182) of all mRNAs showing decreased mRPFs on addition of 500 nM Ded1. Many of the remaining 56% could arise from increased competition for 43S PICs caused by Ded1 stimulation of PIC formation on other mRNAs with long, structured 5′UTRs, which are poorly recruited in the absence of Ded1 but begin competing for PICs when Ded1 is added to the system.

Considering that Ded1 inhibits 48S PIC assembly at the mAUG codons of so few mRNAs (only 182 out of 3052 tested), and that translation initiation continues on at least 46% of these transcripts and merely shifts to iAUGs, our data do not support a model in which a high concentration of Ded1 leads to widespread translational repression by causing mRNAs to undergo phase-transitions into

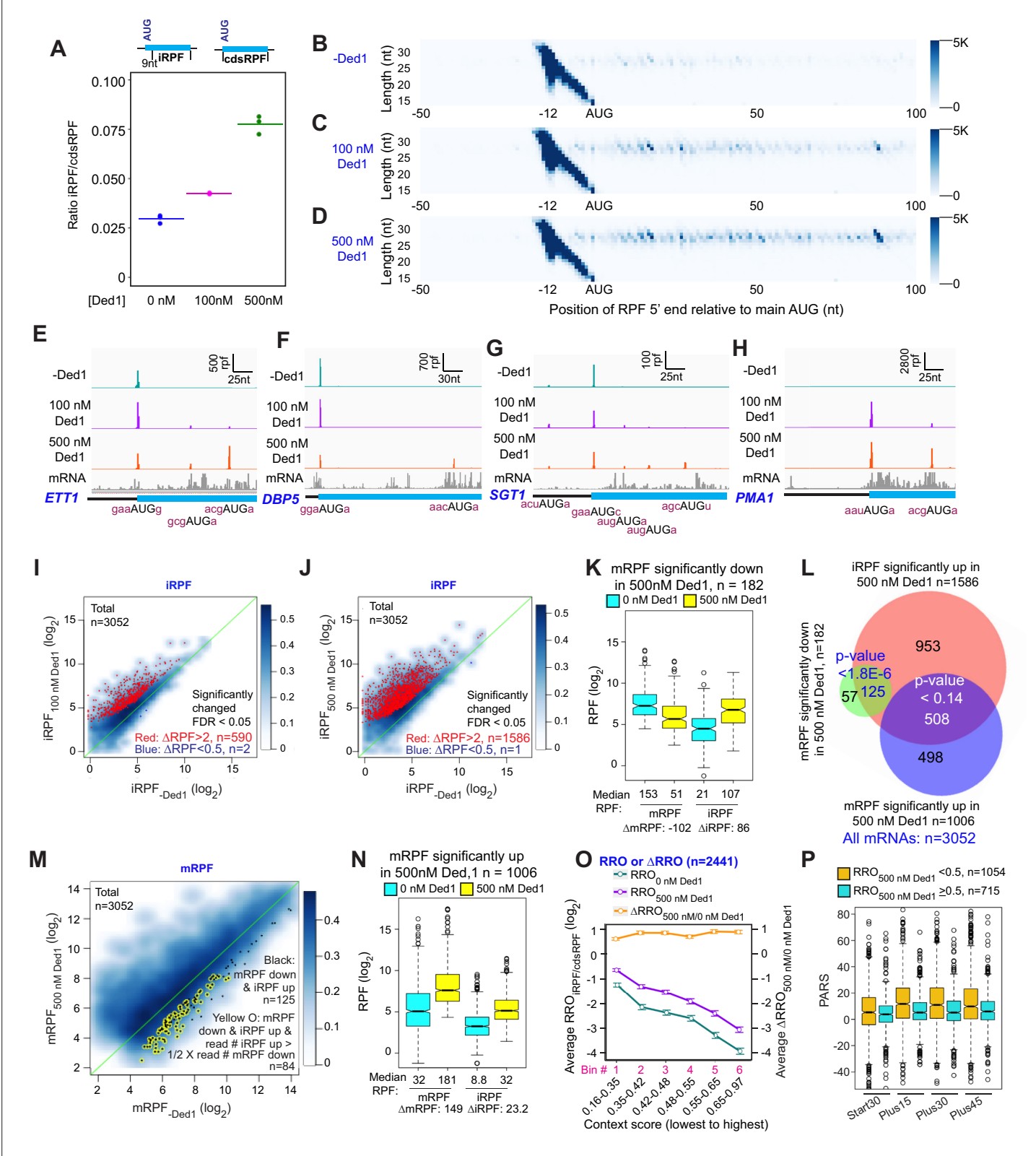

**Figure 5.** Ded1 increases leaky scanning of main start codons. (**A**) The ratio of the total ribosome-protected fragments (RPFs) internal to the coding sequences (CDS) in mRNAs, excluding the main start codon, to the total RPFs for the full CDS including the main start codon (iRPF/cdsRPF ratio) for each of the three replicates at 0, 100, and 500 nM Ded1. iRPFs were counted from the +9 position of the main AUG to the stop codon, while the cdsRPFs were counted from the start codon to the stop codon of the CDS. The average of the three replicates is indicated by the colored bars (red,

*Figure 5 continued on next page*

*Figure 5 continued*

green, and blue for 0, 100, and 500 nM Ded1, respectively). (**B–D**) Metagene plots showing RPF density distribution on all mRNAs aligned to their main AUGs as in *Figure 1B–D* for 0 (**B**), 100 (**C**), and 500 nM Ded1 (**D**), with color scales adjusted to show internal RPFs (iRPFs). (**E–H**) 48S preinitiation complex (PIC) RPFs and input mRNA reads on four selected mRNAs that showed elevated internal ribosome occupancies in 100 and 500 nM Ded1 conditions. The –3 to –1 and +4 context nucleotides surrounding the main AUG, iAUG, or upstream AUG (uAUG) are indicated in brick red text. (**I–J**) Scatterplots comparing $\log_2$ iRPF between 100 (**I**) or 500 nM Ded1 (**J**) to 0 nM Ded1 as described in *Figure 2A and B* for the same group of 3052 mRNAs. Red dots show mRNAs that meet the significance cutoff for increases in iRPF (false discovery rate [FDR] < 0.05, more than twofold increase in iRPFs) and blue dots show mRNAs that meet the significance cutoff for decreases in iRPF (FDR < 0.05, <0.5-fold decrease in iRPFs). (**K**) Boxplot comparing mRPFs and iRPFs between 500 and 0 nM Ded1 for the 182 mRNAs that showed significantly reduced mRPFs with 500 nM Ded1 versus no Ded1. (**L**) Overlaps between mRNAs with significantly elevated iRPFs (orange), significantly reduced mRPFs (green), or significantly increased mRPFs (blue) with 500 nM Ded1 relative to 0 nM Ded1. The Venn diagram was generated and p-values calculated as in *Figure 2D*. (**M**) Scatter plot comparing mRPFs with 500 nM versus 0 nM Ded1. The 125 mRNAs that showed significantly increased iRPFs and significantly decreased mRPFs in 500 nM Ded1 relative to 0 nM Ded1 are labeled by black dots. Among this set, the mRNAs for which the iRPFs increase by at least 50% of the decrease in mRPFs are labeled by yellow circles, to indicate the 84 mRNAs for which the decrease in mRPFs could be responsible for the increase in iRPFs. (**N**) Boxplot comparing mRPFs and iRPFs between 500 nM and 0 nM Ded1 for the 1006 mRNAs that showed significantly increased mRPFs. (**O**) Line plot analysis of average RRO (relative ribosome occupancy; iRPF/cdsRPF ratio; left axis) and $\Delta RRO_{500/0\text{ nM Ded1}}$ (right axis) for 2441 of 3052 mRNAs in (**J**) with 5′UTR length >5 nt binned by main AUG context scores from lowest to highest. (**P**) Boxplot analysis of Start30, Plus15, Plus30, and Plus45 parallel analysis of RNA structure (PARS) scores for mRNAs with $RRO_{iRPF/cdsRPF} < 0.5$ (orange) or ≥0.5 with 500 nM Ded1.

The online version of this article includes the following figure supplement(s) for figure 5:

**Figure supplement 1.** Supporting evidence that Ded1 promotes leaky scanning of the main start codon.

**Figure supplement 2.** mRNAs with reduced recruitment efficiency (RE) at 500 nM Ded1 versus no Ded1 in Rec-Seq assays tend to show increased relative translational efficiency (TE) in the *ded1-cs* mutant versus WT in ribosome profiling experiments.

translationally silenced states. If this is a significant repressive function of Ded1 in vivo, it likely requires even higher concentrations of Ded1, modification of Ded1, or additional factors not present in the Rec-Seq system.

Among the 1586 transcripts showing significantly increased initiation at iAUGs at 500 nM Ded1, roughly approximately one-third (508) exhibit increased rather than decreased mRPFs on Ded1 addition (*Figure 5L*, red/blue intersection). In fact, this trend is evident for the entire group of 1006 mRNAs showing increased PIC assembly at the mAUGs at 500 nM Ded1 (*Figure 5N*), although the median increase in iRPFs conferred by 500 nM Ded1 (23.2 iRPFs) is smaller for this group than that observed for the mRNAs described above where PIC assembly at the mAUG is repressed by Ded1 (86 iRPFs, *Figure 5K*). One possibility is that a high concentration of Ded1 increases readthrough of mAUGs on a large fraction of mRNAs but that in most cases this low-level leaky scanning is offset by larger increases in PIC attachment to mRNAs to yield a net increase in mRPFs despite enhanced mAUG readthrough. Consistent with this proposal, boxplots of RE values show that the mRNAs with mRPFs increased by Ded1 are initiated inefficiently in the absence of Ded1, whereas the mRNAs that display decreased mRPFs in the presence of Ded1 tend to be initiated very efficiently in Ded1's absence (*Figure 5—figure supplement 1B*). Presumably, because the latter mRNAs do not exhibit significantly increased recruitment of 43S PICs when Ded1 is added, the increased leaky scanning of their mAUGs induced by Ded1 is not offset by increased overall recruitment of PICs, resulting in a net decrease in PIC assembly at the mAUGs.

To assess whether the sequence context around the mAUG influences its susceptibility to Ded1-induced leaky scanning, we examined the average fraction of all coding sequence RPFs (cdsRPF) that are iRPFs, a ratio we refer to as relative ribosome occupancy ($RRO_{iRPF/cdsRPF}$), as a function of the context score for the bases surrounding the mAUG. As shown in *Figure 5O*, with both 0 and 500 nM Ded1 the average $RRO_{iRPF/cdsRPF}$ decreases as the context scores around the mAUGs increase (cyan and purple lines), consistent with the idea that the better the context the more stably the PIC is bound and the less likely is leaky scanning through the mAUG. Plotting the ratio of RRO values at 500 nM to 0 nM Ded1 yields a flat line with $\log_2\Delta RRO$ values of ~0.75 for all bins (*Figure 5O*, orange line). These last results suggest that addition of 500 nM Ded1 increases the frequency of leaky scanning by ~1.7-fold regardless of mAUG context. This in turn implies that Ded1 does not inspect the context sequence but rather promotes scanning to reduce the dwell time of the PIC at the AUG similarly for both weak and strong contexts.

Rather than invoking leaky scanning, it could be proposed that the RPFs formed at iAUGs originate from a small fraction of mRNA isoforms with transcription initiation sites within the CDSs (*Arribere*

*and Gilbert, 2013*; *Lu and Lin, 2019*) that contain the internal initiation sites as the first AUGs encountered on scanning from the cap. Alternatively, the iRPFs could be formed on uncapped mRNAs cleaved at the 5' end in cells or in vitro, which would require cap-independent initiation. In either case, Ded1 could stimulate PIC assembly at the iAUGs by the same mechanism identified above for canonical mAUGs. Several lines of evidence argue against this possibility and instead favor the Ded1-induced leaky scanning model proposed above. First, the level of iAUG initiation relative to mAUG initiation (RRO) depends strongly on the sequence context surrounding the mAUG (*Figure 5O*), as expected for the leaky scanning model but inconsistent with a model in which iAUG occupancy arises from 5'-truncated mRNA isoforms or fragments. In addition, mRNAs that have a high level of iAUG occupancy relative to mAUG occupancy (RRO ≥ 0.5) tend to have less secondary structure in the 5' ends of their coding sequences than do mRNAs with less iAUG occupancy (*Figure 5P*, cols. 3–8, compare blue and orange boxes), consistent with a model in which secondary structures just downstream of the mAUG inhibit further scanning and diminish mAUG readthrough (*Kozak, 1990*). This result is not predicted by the truncated mRNA isoform/fragment model. Finally, it is difficult to account for the reciprocal effects of Ded1 in repressing mRPFs while inducing iRPFs by comparable amounts for the same genes (*Figure 5K*) if the events occur independently on different transcript isoforms.

Overall, these data suggest that a high Ded1 concentration increases leaky scanning of the mAUG for approximately half (1586/3052; *Figure 5J*) of the mRNAs in the yeast translatome that were observed in this study. In some cases, this leads to a decrease in mRPFs and may account for nearly 50% of the mRNAs for which Ded1 suppresses 48S PIC assembly at the mAUG, while in others the overall enhancement of PIC loading on the mRNA induced by Ded1 likely offsets the increased leaky scanning for a net stimulation of 48S PIC assembly at the mAUG. The inhibitory effect of Ded1 on proper initiation at mAUGs for the subsets of mRNAs described above for which mRPFs are significantly decreased by 500 nM Ded1 in the Rec-Seq experiments also appears to be operative in vivo as the median TEs of these sets of mRNAs are increased in *ded1-cs* cells at the non-permissive temperature relative to WT cells (*Figure 5—figure supplement 2*), signifying Ded1-hypodependence. These data are consistent with Ded1 activity diminishing the relative translation levels of these mRNAs in WT cells, in alignment with our observations in the Rec-Seq experiments.

## Ded1 does not generally increase initiation at canonical AUG codons in Rec-Seq by suppressing alternative initiation events in 5′UTRs

It has been proposed that a key function of Ded1 is to increase the fidelity of start codon selection by promoting readthrough of upstream AUG (uAUG) and near-cognate codons to boost the fraction of scanning PICs that initiate at the mAUG codon (*Guenther et al., 2018*). To determine whether this mechanism operates in the reconstituted system, we analyzed the effect of Ded1 in Rec-Seq reactions on the total RPFs mapping to the 5′-UTRs of mRNAs (uRPFs). We observed that 40–60% of all uRPFs detected when Ded1 is present in the reaction mapped to the *GCN4* 5′UTR (*Figure 6—figure supplement 1*), and we therefore excluded *GCN4* mRNA in calculating the [total uRPF]/[total mRPF] ratio in order to measure effects on the overall translatome. In the absence of Ded1, the ratio of total uRPFs to mRPFs was 0.0021 (*Figure 6A*, col. 1). Addition of 100 nM Ded1 decreased the ratio to 0.0017 and increasing the concentration to 500 nM decreased it further only slightly, to 0.0016 (*Figure 6A*, cols. 2–3). These values are similar to the 0.0027 value measured for the ratio of uRPFs to mRPFs in WT yeast cells in a previous ribosome profiling study of 5′UTR translation (*Kulkarni et al., 2019*), suggesting that the level of 5′UTR translation occurring in the in vitro Rec-Seq system is within twofold of that taking place in vivo. A decreased uRPF/mRPF ratio on addition of Ded1 is expected if Ded1 suppresses PIC assembly at upstream start codons as proposed in the aforementioned model. However, even in the absence of Ded1, uRPFs represent a very small fraction of all RPFs and this fraction is decreased by only ~30% upon addition of Ded1.

We next visualized the effects of Ded1 on upstream initiation in individual mRNAs by plotting the uRPFs in the absence of Ded1 versus the presence of 100 or 500 nM Ded1 (*Figure 6B and C*, respectively). Going from 0 to 100 nM Ded1 led to a significant reduction (ΔRPFs < 0.5, FDR < 0.05) in uRPFs for only nine mRNAs (*Figure 6B*, dark blue dots). Moreover, only one mRNA (*SXM1*) showed a decrease in uRPFs comparable in magnitude (>50%) to the corresponding increase in mRPFs in the manner expected if Ded1-stimulated leaky scanning of uAUGs stimulates PIC assembly at the mAUG (*Figure 6B*, yellow circle). In contrast, uRPFs for 14 mRNAs were significantly increased (ΔRPFs > 2,

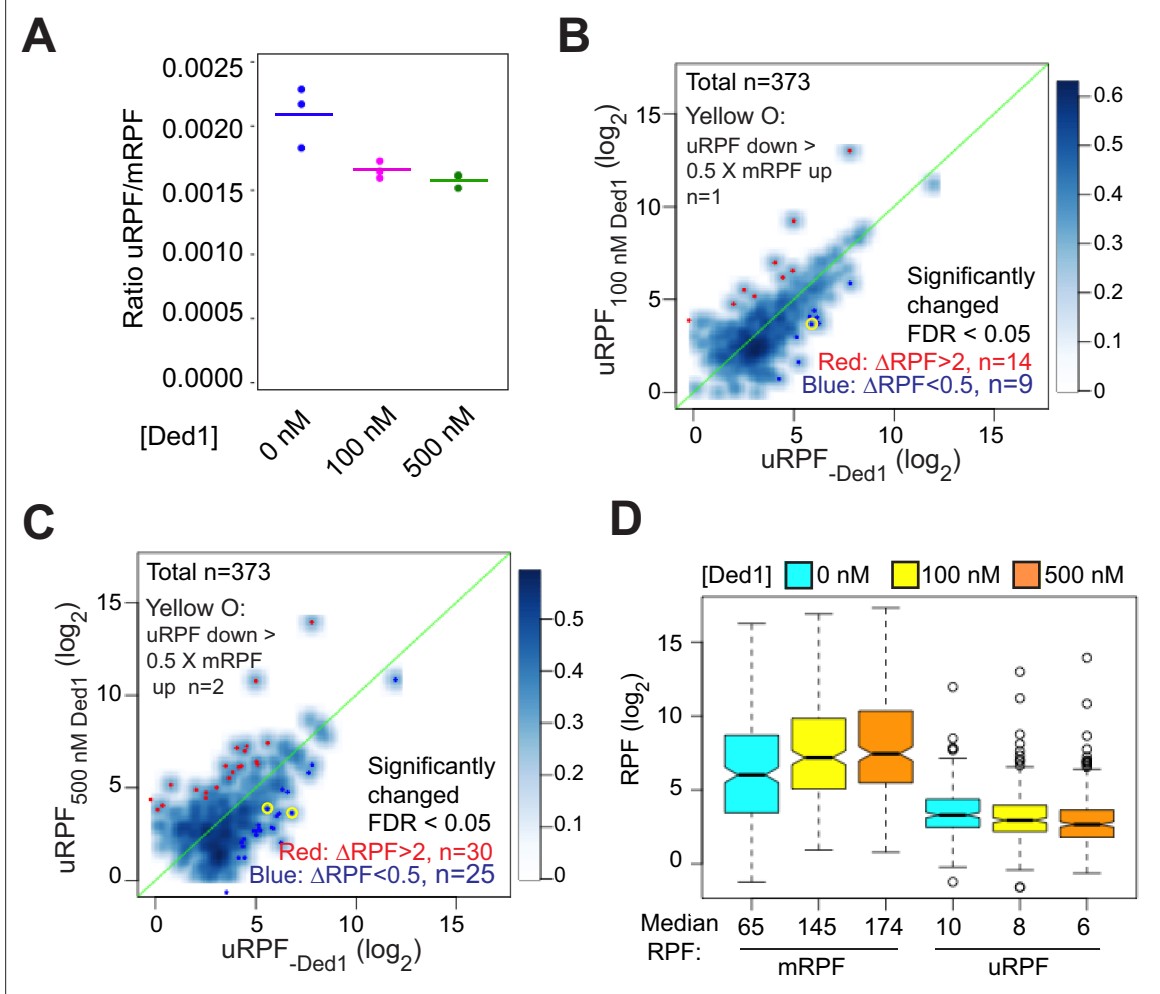

**Figure 6.** Ded1 modestly promotes readthrough of start codons in 5'UTRs of mRNAs. (**A**) The ratios of ribosome-protected fragments (RPFs) in 5'UTRs to RPFs on main start codons (uRPF/mRPF ratios) for each of the three replicates with 0 (blue), 100 (magenta), or 500 (green) nM Ded1. The 5'UTR RPFs (uRPFs) were counted from the 5' end of the mRNA to the –5 position relative to the main AUG. The mean of the three replicates is indicated by the colored bars. uRPFs on *GCN4* mRNA were analyzed separately (**Figure 6—figure supplement 1**) because 42 and 57% of all uRPFs were mapped to the *GCN4* 5'UTR in assays with 100 and 500 nM Ded1, respectively. (**B, C**) Scatterplots comparing uRPFs in the presence of either 100 nM (**B**) or 500 nM (**C**) Ded1 versus 0 nM Ded1. mRNAs with significantly increased or decreased uRPFs in the presence of Ded1 are indicated in red or dark blue dots, respectively. The criteria used for significance were false discovery rate (FDR) < 0.05 and a more than twofold increase or decrease. Yellow circles denote the very few mRNAs whose uRPF read number decreases by more than 50% of the increases in their mRPF reads, indicating a clear reciprocal relationship between the decrease in uRPFs and the increase in mRPFs. (**D**) Boxplot analysis of mRPF and uRPF read numbers for the 257 mRNAs that had both uRPFs and mRPFs ≥2 reads averaged over all assays conducted at 0, 100, and 500 nM Ded1 concentrations. Unlogged median RPF numbers are labeled under each box.

The online version of this article includes the following figure supplement(s) for figure 6:

**Figure supplement 1.** Approximately half of all uRPFs map to the *GCN4* 5'UTR.

FDR < 0.05), presumably due to enhanced PIC attachment to these mRNAs caused by addition of Ded1. Similar results were obtained in going from 0 to 500 nM Ded1, which resulted in only 25 mRNAs with significantly decreased uRPFs, and only two showing reductions >50% of the magnitude of the observed increase in mRPFs for the mRNA (*MSC7* and *UIP5*), whereas 30 mRNAs showed significant increases in uRPFs (**Figure 6C**). In addition, there are at least as many cases in which upstream initiation is increased rather than decreased, most likely due to overall enhancement of PIC attachment to these mRNAs by Ded1. These data indicate that under the conditions of the Rec-seq experiments, Ded1 directly increases PIC attachment to mRNAs and scanning to the main start codons rather than primarily functioning by diminishing inhibitory initiation in 5'UTRs.

Additional evidence supporting our conclusion came from comparing the magnitude of changes in uRPFs versus mRPFs conferred by Ded1 for a group of 257 mRNAs containing RPFs in both 5′UTRs and at mAUG codons exceeding a minimum threshold read abundance. Overall, the mRPFs are an order of magnitude higher than uRPFs in the absence or presence of Ded1 at either concentration (*Figure 6D*, see median values below each column). Addition of 100 nM Ded1 decreases the median uRPFs by 2, from 10 to 8, but increases mRPFs by 80, from 65 to 145. Likewise, with 500 nM Ded1 relative to no Ded1, uRPFs decrease by 4, whereas mRPFs increase by 109. Thus, the magnitude of the RPF decreases in 5′UTRs are much too small to account for the increases in RPFs at mAUG codons elicited by addition of Ded1. Taken together, our data indicate that, at least in the reconstituted in vitro system, Ded1 functions by directly increasing PIC attachment to mRNAs and scanning to the main start codons rather than by diminishing inhibitory initiation in 5′UTRs.

## eIF4A plays a distinct role from Ded1 and stimulates recruitment of most mRNAs regardless of their 5′UTR lengths or structures

Previous work in vitro and in vivo has indicated that eIF4A and Ded1 play distinct roles in promoting translation initiation. In the reconstituted system, no assembly of 48S PICs was observed for a variety of individual mRNAs in the absence of eIF4A but presence of Ded1, indicating that Ded1 cannot take the place of eIF4A (*Gupta et al., 2018*). In contrast, 48S PIC formation occurred efficiently on many mRNAs in vitro in the presence of eIF4A but absence of Ded1, and all mRNAs studied required eIF4A for PIC assembly, regardless of their degree of secondary structure (*Pestova and Kolupaeva, 2002*; *Yourik et al., 2017*; *Gupta et al., 2018*). In vivo ribosome profiling studies also led to the conclusion that eIF4A is universally required for translation of most mRNAs, whereas Ded1 preferentially stimulates translation of mRNAs with long, structured 5′UTRs (*Sen et al., 2015*).

Our results here strongly support the previous conclusions that Ded1 functions by alleviating structural impediments to PIC loading and scanning in the 5′UTRs of particular mRNAs burdened by these features. To probe further functional differences between Ded1 and eIF4A, we performed Rec-Seq reactions in the absence of Ded1 and presence of either 5000 nM (1×) or 500 nM (0.1×) eIF4A. It was not possible to do the experiment in the absence of eIF4A because the factor is essential for 48S PIC formation on most, if not all, mRNAs. A scatterplot of mRPFs at 5000 versus 500 nM eIF4A shows that PIC assembly at the mAUGs of most mRNAs increases when the concentration of eIF4A is raised tenfold (*Figure 7A*, light blue density above the diagonal). Of the 2809 mRNAs observed in these experiments, 782 reached the level of significance in DEseq2 analysis of ≥2-fold change in mRPFs and FDR < 0.05. Only one mRNA (*SCW10*) met the criteria for significantly decreased mRPFs (*Figure 7A*, dark blue dot). Consistent with these results, increasing the eIF4A concentration from 500 to 5000 nM increased the REs of almost all mRNAs, with a median change of ~5.7-fold (*Figure 7B*, col. 1). By comparison, addition of either 100 or 500 nM Ded1 in a background of 5000 nM eIF4A increased the median RE by only 1.3-fold, with RE for many mRNAs not changing (*Figure 7B*, cols. 2–3). Rather, a large number of outliers are increased dramatically more than the median ΔRE value when Ded1 is added to the reactions, consistent with the preferential stimulation of 48S PIC assembly by Ded1 for mRNAs with long, structured 5′UTRs shown above (*Figure 3*). In our previous Ribo-Seq experiments on an eIF4A mutant, where changes in TE were determined relative to the effect on the average mRNA, the vast majority of mRNAs displayed no change in relative TE despite a strong reduction in bulk translation, implying TE reductions of similar magnitude for nearly all mRNAs (*Sen et al., 2015*).

To examine whether length or structure of 5′UTRs influences the stimulatory effects of eIF4A in Rec-Seq, we plotted ΔRE values between 500 and 5000 nM eIF4A as a function of 5′UTR length or Max30 PARS values. Importantly, the median ΔRE values remain constant across the first five bins of 5′UTR lengths and four bins of Max30 PARS scores and actually decrease significantly in the sixth bins (*Figure 7C and D*). This behavior contrasts with that described above for addition of Ded1 to Rec-Seq reactions, which showed progressively greater enhancement of RE as 5′UTR length and Max30 PARS scores increased (*Figure 3B, C, F, and G*). The distinct effects of increasing eIF4A concentration versus addition of Ded1 on mRNAs binned according to 5′UTR length, 5′UTR Max30 PARS, or 5′UTR total PARS values are depicted in *Figure 7E–G*. Consistent with these results, inspection of RPF traces for individual mRNAs reveals that eIF4A enhances 48S PIC assembly on mRNAs judged to be either hypo- or hyperdependent on Ded1 in Ribo-Seq experiments (*Sen et al., 2015*; *Figure 7H–K*). Overall, these results are consistent with previous conclusions that eIF4A stimulates recruitment of all mRNAs,

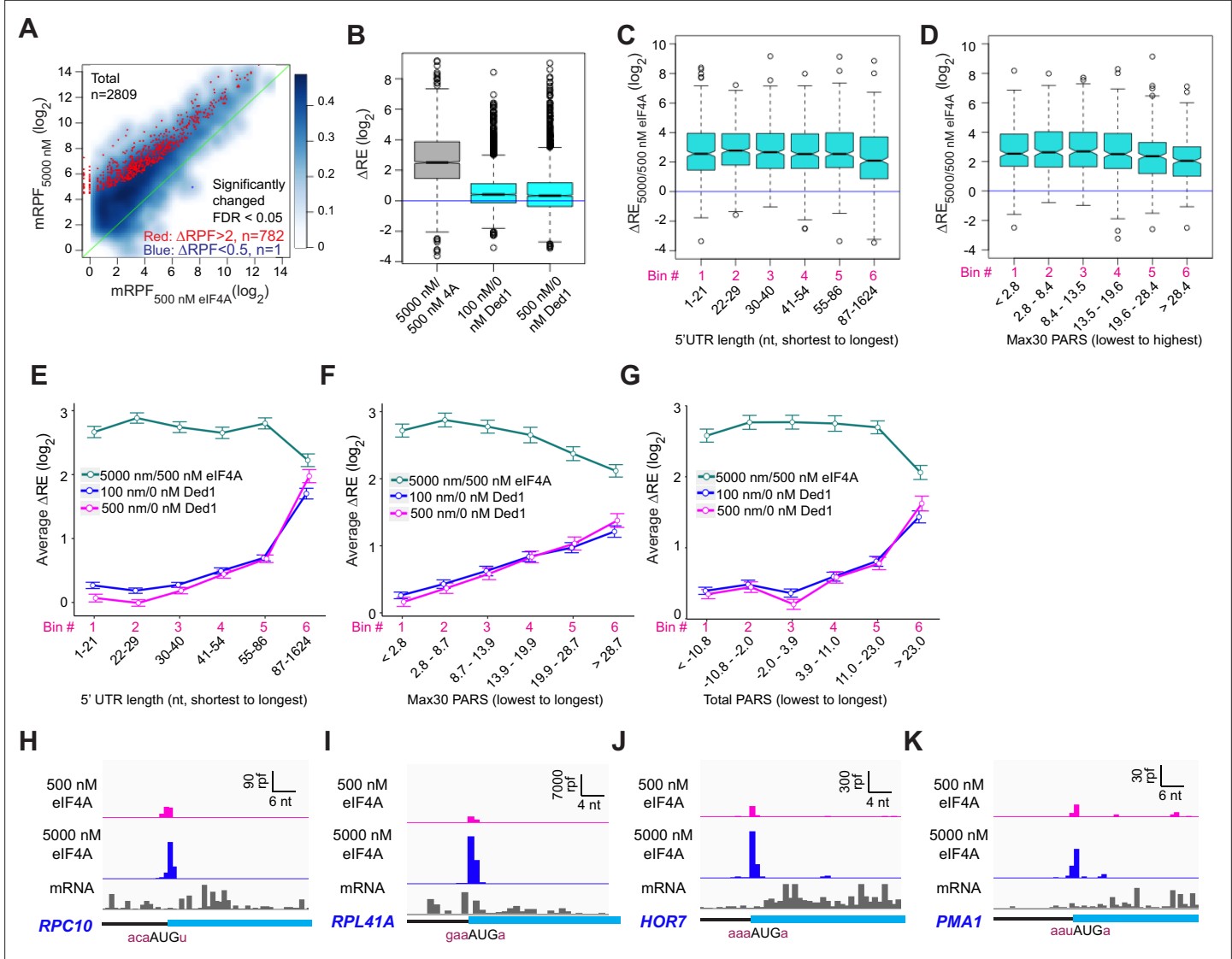

**Figure 7.** eIF4A enhances the recruitment of almost all mRNAs. (**A**) Scatterplots of normalized read densities mapped to main AUGs (mRPFs) with 5000 nM versus 500 nM eIF4A for the 2809 mRNAs with ≥8 total reads in four samples (two replicates each for 5000 and 500 nM eIF4A). Red and blue dots show mRNAs with mRPFs significantly increased or decreased, respectively, at 5000 versus 500 nM eIF4A. The criteria for significance were false discovery rate (FDR) < 0.05 and a more than twofold change in ribosome-protected fragments (RPFs). (**B**) Boxplot comparing $\Delta RE_{5000/500\ nM\ eIF4A}$ (grey) to $\Delta RE_{100/0\ nM\ Ded1}$ and $\Delta RE_{500/0\ nM\ Ded1}$ (cyan) for the 2698 mRNAs that passed the cutoff mentioned in (**A**) in both experiments. (**C**) Boxplot of $\Delta RE_{5000/500\ nM\ eIF4A}$ for the 2538 of all 2809 mRNAs in (**A**) that have annotated 5'UTRs, divided into six equal-sized bins according to 5'UTR lengths from the shortest to the longest. (**D**) Similar to (**C**), but for the 1708 of all 2809 mRNAs in (**A**) with annotated parallel analysis of RNA structure (PARS) scores, binned by Max30 PARS from the lowest to the highest. (**E–G**) Line plots of mean $\log_2$ RE changes, $\Delta RE_{5000/500\ nM\ eIF4A}$ (green), $\Delta RE_{100/0\ nM\ Ded1}$ (magenta), and $\Delta RE_{500/0\ nM\ Ded1}$ (blue), for mRNAs divided into six equal-sized bins according to the specific mRNA features of 5'UTR length (**E**), Max30 PARS score (**F**), or total PARS score (**G**). (**H–K**) RPFs assembled with 500 or 5000 nM eIF4A and input mRNA reads for mRNAs shown previously to be hypodependent on Ded1 (*RPC10*, *RPL41A*, *HOR7*) or hyperdependent on Ded1(*PMA1*) by in vivo ribosome profiling experiments in the *ded1-cs* mutant.

regardless of their degree of secondary structure, whereas Ded1 specifically acts on mRNAs with long, structured 5'UTRs. Our data are also in line with previous work indicating that Ded1 has much stronger helicase activity than eIF4A (*Rogers et al., 1999*; *Yang et al., 2007*; *Rajagopal et al., 2012*) and thus the former is likely to play a role in unwinding stable secondary structures whereas the latter mediates engagement of mRNAs with the 43S PIC by resolving ensembles of weaker interactions within mRNAs or by modulating the structure of the mRNA channel of the 40S ribosomal subunit (*Sokabe and Fraser, 2017*; *Yourik et al., 2017*).

## Discussion

We have developed a deep sequencing-based approach, 'Rec-Seq', for measuring the efficiency with which 48S PICs form on each mRNA in the yeast transcriptome in a reconstituted in vitro system. Using this approach, we have provided evidence that the DEAD-box translation initiation factor Ded1 specifically stimulates 48S PIC formation on mRNAs with long, structured 5′UTRs, which supports the model that the factor generally acts by unwinding secondary structures in 5′UTRs to promote 43S PIC binding and scanning to locate the start (*Berthelot et al., 2004*; *Sen et al., 2015*; *Guenther et al., 2018*; *Gupta et al., 2018*; *Sen et al., 2019*; *Sen et al., 2021*). We showed that 48S PIC formation on a set of ~1000 mRNAs is significantly less efficient in the absence of Ded1 than in its presence, demonstrating a positive function for the factor on these mRNAs. It is striking that the Ded1-stimulated mRNAs identified in our in vitro system include the great majority of mRNAs identified as being hyperdependent on Ded1 in vivo by ribosome profiling of a *ded1-cs* mutant. This concordance argues that the mRNAs showing the strongest dependence on Ded1 for efficient translation in vivo are stimulated by the factor at the stage of 48S PIC assembly. It further argues against the possibility that the TE reductions observed for Ded1 hyperdependent mRNAs in *ded1-cs* cells frequently result from dominant inhibitory properties of the mutant *ded1* proteins. Ded1 also stimulated 48S PIC assembly in Rec-Seq for many mRNAs not classified as Ded1-hyperdependent by Ribo-Seq. However, as explained above, translation of these mRNAs was likely impaired by the *ded1-cs* mutation but to a degree that was less than the twofold cutoff required for statistical significance or that was less than the effect on the average mRNA and thus yielded a positive change in the relative TE values determined in the Ribo-Seq analysis.

Previous studies have indicated that yeast Ded1 and its mammalian ortholog DDX3 can stimulate the subunit joining step of translation initiation (*Senissar et al., 2014*; *Wang et al., 2022*). Although our Rec-Seq system only monitored steps up to 48S PIC formation and was not able to probe effects on 60S subunit joining, it should be noted that nearly 90% of the Ded1-hyperdependent mRNAs identified by Ribo-seq analysis of the *ded1-cs* mutant were also identified in our in vitro Rec-Seq experiments monitoring only 48S PIC assembly. This result argues strongly that Ded1 affects 48S PIC assembly both in vitro and in vivo, although it does not rule out an additional effect of the factor on subunit joining.

Compared to the sizeable group of >1000 mRNAs for which 500 nM Ded1 stimulated 48S PIC assembly, we identified a much smaller set of only 182 mRNAs for which 48S PIC assembly was significantly repressed by Ded1 in Rec-Seq experiments. We obtained evidence that at least 84 of these transcripts were repressed owing to Ded1-stimulated leaky scanning of the mAUG codon rather than by Ded1 blocking PIC attachment or scanning. As a group, the mRNAs exhibiting Ded1 repression in Rec-Seq show increased relative TE in Ribo-Seq analysis of the *ded1-cs* mutant, regardless of whether they display evidence of Ded1-enhanced leaky scanning of the mAUG codons in Rec-Seq (*Figure 5—figure supplement 2*). This behavior is consistent with loss of Ded1 repression or with smaller than average reductions in TE in *ded1-cs* versus WT cells. The latter possibility is consistent with our finding that the 182 mRNAs inhibited by 500 nM Ded1 in Rec-Seq assemble 48S PICs very efficiently without Ded1 (*Figure 5—figure supplement 1*). Being relatively independent of Ded1 for PIC assembly, these mRNAs should experience a smaller than average reduction in translation in *ded1-cs* cells owing to reduced competition for limiting 43S PICs with the Ded1-dependent mRNAs that should be strongly impaired for PIC assembly. We argued above that a similar mechanism can also explain the repression of RE by Ded1 in Rec-Seq experiments for the subset of 98 mRNAs without evidence of appreciable leaky scanning of the mAUG codon, owing to increased competition for 43S PICs with the Ded1-stimulated transcripts. It remains to be seen whether Ded1 acts directly to repress 48S PIC assembly on any individual mRNAs in yeast. It is possible that at higher concentrations of Ded1 than were achievable in these in vitro experiments or in the presence of additional factors that modify Ded1's ATPase or RNA binding activities the factor could directly inhibit a subset of mRNAs by acting as an mRNA clamp that impedes scanning by the PIC or by sequestering the mRNAs in insoluble condensates. It might be interesting in the future to test candidate factors in Rec-Seq to determine if they switch Ded1 from being a stimulatory helicase to an inhibitory mRNA clamp that removes transcripts from the soluble phase.

We found that the presence of 500 nM Ded1 in Rec-Seq reactions increased 48S PIC assembly at internal AUG codons on ~50% of all detected transcripts. In approximately one-third of these 1586

transcripts, there is a parallel increase in RPFs at the mAUG codons, suggesting that Ded1 stimulates 43S PIC attachment at the 5′ ends of these transcripts to increase initiation proportionately at all AUGs in the mRNA. As mentioned, in ~5% of the mRNAs (84/1586), the increased internal initiation may result from leaky scanning of the mAUG codons, as we found a comparable reduction in initiation at the mAUG. In this view, Ded1 might resolve secondary structures positioned just downstream of the mAUG codon that increase the dwell time of the PIC and increase the probability of PIC assembly versus continued scanning downstream that leads to initiation on iAUG codons, in the manner first described for mammals by *Kozak, 1990*. Interestingly, Ded1 enhanced this apparent mAUG read-through to similar extents regardless of the mAUG context, which would imply that Ded1 promotes scanning of the PIC past the mAUG without inspecting the context nucleotides surrounding it.

Ded1 also significantly reduced RPFs at upstream AUG and near-cognate start codons in the 5′UTRs of a small number of transcripts (25 mRNAs at 500 nM Ded1), possibly by the same mechanism mentioned above involving unwinding of downstream secondary structures. However, this reduction was associated with a comparable increase in RPFs at the mAUG for only two of these mRNAs. Our findings that PIC assembly at uAUGs in the absence of Ded1 is rare, and that suppression of these rare occurrences by Ded1 cannot explain the much larger increases in initiation at the mAUGs conferred by Ded1, argue that, at least in this in vitro system, Ded1 functions primarily by directly promoting 43S PIC attachment to mRNAs and scanning to the main start codons rather than by diminishing inhibitory initiation in 5′UTRs.

Finally, our data show that Ded1 and eIF4A, another essential DEAD-box translation initiation factor, have distinct functions. Unlike Ded1, which has pronounced specificity for activating initiation on mRNAs with long, structured 5′UTRs, eIF4A strongly promotes initiation on nearly all mRNAs, regardless of the length or degree of structure of their 5′ leaders. Because our system employs internal 'spike-in' standards, we were able to measure the absolute enhancement provided by eIF4A for mRNA recruitment upon increasing its concentration from 500 to 5000 nM, a 5.7-fold increase in median RE (*Figure 7B*). In contrast, addition of Ded1 to the system produced a much smaller change in median RE, of only 1.3-fold, while conferring much larger effects, >10-fold, on many mRNAs with long, structured 5′UTRs. These findings support our previous conclusions from Ribo-Seq analysis of Ded1 and eIF4A mutants that Ded1 preferentially stimulates translation of mRNAs burdened with structured 5′UTRs while eIF4A enhances the translation of nearly all mRNAs equally (*Sen et al., 2015*).

Our data provide information on the intrinsic efficiency of 48S PIC formation on cellular mRNAs in the yeast translatome. In the presence of 500 nM Ded1, RE values follow a roughly normal distribution spanning a 1000-fold range (*Figure 2H*), indicating that even in the presence of Ded1, there are still large differences in the intrinsic efficiencies with which mRNAs are recruited to the 43S PIC and scanned to locate the start codon. In the presence of Ded1, there is little dependence of RE on 5′UTR structure or length (*Figure 3A and E*) and thus other mRNA characteristics must set the 48S PIC formation efficiencies. The strength of sequence context around the start codon has a modest (approximately twofold) effect (*Figure 3—figure supplement 2A*), but not nearly enough to explain the range of REs observed. The 1000-fold range we observe here is strikingly similar to the range reported in a systematic study of 5′UTR variants in a yeast lysate-based translation system (*Niederer et al., 2022*). In that work, the authors provided evidence that a variety of sequence elements in 5′UTRs can enhance or inhibit translation initiation through a range of mechanisms. Thus, it is likely that in the presence of Ded1 no single mRNA feature sets the efficiency of 48S complex assembly and instead a multitude of effects and interactions are involved, possibly including specific interactions between mRNA elements and initiation factors or the ribosome (*Niederer et al., 2022*). In addition, mRNA elements outside of the 5′UTR, including the poly(A) tail and its interaction with PABP, might influence the efficiency of 48S PIC formation. Further studies will be required to elucidate fundamental principles and specific cases of how mRNA sequences dictate TE.

## Study limitations

As with all studies, this study has caveats and limitations. The fidelity of start codon recognition in the in vitro Rec-Seq system might differ from what occurs in vivo. Likewise, the use of an unmodified, in vitro transcribed initiator tRNA might lead to altered patterns of start codon recognition in the Rec-Seq system relative to what is observed within living cells. Previous studies of start codon recognition in the in vitro reconstituted yeast translation initiation system indicated that the general trend of start

codon preference reflects what is expected from in vivo studies, with AUG codons leading to the most stable PICs, followed by near-cognate codons with first position mismatches (*Kolitz et al., 2009*). The Rec-Seq system shows a very strong preference for AUG codons over near-cognates (e.g., *Figures 1 and 6*) and for good Kozak consensus sequences surrounding the start codon (*Figure 5O*). Nonetheless, it will be worthwhile in future studies in the Rec-Seq system to explore the effects of buffer conditions, temperature, and tRNA modifications on start codon recognition fidelity and its interplay with initiation factors and mRNA structure.

The current version of the Rec-Seq system isolates events up to and including formation of a stable 48S PIC on an mRNA start codon, but does not allow these PICs to progress to 80S initiation complexes via 60S subunit joining. It is possible that subunit joining and 80S complex formation could alter some of the patterns we have observed here. For example, rapid subunit joining could diminish the Ded1-induced leaky scanning we have observed by stabilizing the complexes and making them resistant to displacement and further scanning. Further studies will be required to assess this possibility as well as any potential effects of Ded1 on the subunit joining step (*Senissar et al., 2014*; *Wang et al., 2022*). It should also be noted that the Rec-Seq system does not disentangle the steps of PIC binding to mRNA, scanning, and start codon recognition from one another and thus independent effects on each of these processes cannot currently be assessed with the assay.

Finally, in these experiments we monitored endpoints for 48S PIC formation at a single timepoint (15 min) under single-turnover conditions in which neither multiple rounds of initiation on a single start codon nor ribosome reinitiation events would be observed. The use of single timepoints rather than kinetic measurements likely obscured some time-dependent effects. Future experiments in which multiple timepoints are taken might allow us to observe changes in the kinetics of 48S PIC formation upon addition of Ded1 or other factors to the system. The fact that we cannot observe multiple-turnover initiation or reinitiation events using this system could limit its utility in studying translational control events that require these processes.

# Materials and methods
## Purification of yeast total mRNA

WT *Saccharomyces cerevisiae* strain F729/BY4741 (*MATa his3Δ1 leu2Δ0 met15Δ0 ura3Δ0*) (*Winzeler et al., 1999*) was cultured in Yeast Extract–Peptone–Dextrose (YPD) medium to $OD_{600}$ of ~1. Harvested cells were washed once with cold water and stored at –80°C before use. The frozen cell pellet from a 500 ml culture was thawed on ice and mechanically disrupted by vortexing with glass beads three times for 2 min each at maximum speed in a cold room (4°C) in 2 ml of buffer RL (provided in the kit), 3 ml phenol:chloroform:isoamyl alcohol 25:24:1 (pH 5.2) ,and 3 ml ice-cold glass beads. Following 5 min incubation at 65°C and 5 min centrifugation at 14,000 rpm in a Sorvall LYNX 6000 Superspeed centrifuge using a Fiberlite F14-14 × 50 cy rotor, the supernatant was transferred to a fresh tube and total RNA was prepared using a GenElute total RNA purification Maxi kit (Sigma-Aldrich; RNB200) following the manufacturer's protocol. 400 µg total RNA in 250 µl water was applied to a GenElute mRNA miniprep kit (Sigma-Aldrich; MRN70-1KT), following the manufacturer's protocol. The purified total mRNA was treated by 5'-Phosphate-Dependent Exonuclease (Lucigen, TER51020) to degrade RNAs with 5' monophosphates, such as 18S and 25S ribosomal RNAs, at 0.3 µg/µl mRNA in 1× buffer A (provided in the kit), 1 U/µl RiboGuard RNase Inhibitor (Lucigen; RG90925), 0.1 U/µl 5'-Phosphate-Dependent Exonuclease. Following a 1 hr incubation at 30°C, the total mRNA was extracted using phenol:chloroform:isoamyl alcohol 25:24:1 (pH 5.2), ethanol precipitated, resuspended in 10 mM Tris (pH 8.0), and stored at –80°C. The mass concentration of purified total mRNA was determined using the RNA concentration function on a NanoDrop One (Thermo Fisher). The mass concentration was then converted to a molar concentration using an online tool 'Weight to Molar Quantity (for nucleic acids) converter' (https://www.bioline.com/media/calculator/01_07.html), using the average molecular weight of nucleoside monophosphates and assuming the average length of mRNAs is 1000 nt (*Miura et al., 2008*). Also, 75% of mRNAs (2719 of 3640) not observed in the Rec-Seq analysis had densities below the median (2.3 reads per nucleotide).

## Purification of 40S ribosome subunits and translation initiation factors

Eukaryotic initiation factors eIF1, eIF1A, eIF2, eIF3, eIF4A, eIF4B, eIF4G·4E, and eIF5 were expressed and purified as described previously (*Acker et al., 2007*; *Mitchell et al., 2010*; *Rajagopal et al., 2012*; *Gupta et al., 2018*). 40S ribosomal subunits were prepared as described in *Munoz et al., 2017*. Ded1 protein (N-terminal His6-tag, pET22b vector) was purified as described previously (*Gupta et al., 2018*).

## Preparation of spike-in mRNAs and charged initiator tRNA

The templates for in vitro transcription of the 5′ fragments of *FLUC* and *RLUC* genes were amplified by PCR from plasmid FJZ1061 (*Zhou et al., 2020*) using primers T7-FLUC and FLUC-R for *FLUC*, T7-RLUC and RLUC-R for *RLUC* (*Table 1*). The mRNAs and initiator tRNA were transcribed by run-off transcription using T7 RNA polymerase and gel purified as described previously (*Acker et al., 2007*; *Mitchell et al., 2010*). mRNAs were capped (m7GpppG) using GTP and vaccinia virus capping enzyme (*Mitchell et al., 2010*). Initiator tRNA was methionylated in vitro using methionine and *Escherichia coli* methionyl-tRNA synthetase as previously described (*Walker and Fredrick, 2008*; *Yourik et al., 2017*).

## mRNA recruitment assays on spike-in mRNAs and total mRNA

48S PICs were assembled on pre-mixed spike-in mRNAs (molar ratio FLUC/RLUC ~1:10) as described previously (*Mitchell et al., 2010*; *Yourik et al., 2017*) in 1× Recon buffer (30 mM HEPES-KOH, pH 7.4, 100 mM KOAc, 3 mM Mg(OAc)$_2$, and 2 mM DTT) containing 300 nM eIF2, 0.5 mM GDPNP·Mg$^{2+}$, 200 nM Met-tRNA$_i$Met, 1 µM eIF1, 1 µM eIF1A, 300 nM eIF5, 300 nM eIF4B, 300 nM eIF3, 30 nM 40S subunits, 5 µM eIF4A, 30 nM eIF4E·eIFG, 5 µM Ded1, and 15 nM mRNA. The reaction was incubated at 26°C for 20 min before being rapidly quenched by adding stop buffer (1× Recon buffer, 37.5 mM glucose and 0.04 U/µl Hexokinase [Roche Diagnostics, REF:11426362001]) in a 3:10 v/v ratio. RPFs of 48S PICs were generated by incubating the reaction with RNase I (Thermo Fisher; AM2295) at a final concentration of 2.5 U/µl, followed by adding SUPERaseIN RNase inhibitor (Thermo Fisher; AM2696) at a final concentration of 1.2 U/µl. Recruitment reactions using purified yeast total mRNA with different Ded1 concentrations were performed similarly as for spike-in mRNAs, but incubated at 22°C for 15 min, with all of the same initiation factor concentrations except 60 nM input total mRNA, 120 nM eIF4E.eIFG and three different Ded1 concentrations (0 nM, 100 nM, and 500 nM), with three replicates for each condition. The recruitment reactions were rapidly quenched by stop buffer and incubated with RNase I to generate RPFs from 48S PICs as described above. Aliquots (4.5 µl) of spike-in RPFs were mixed with the experimental sample RPFs and resolved on a 5–25% sucrose gradient by ultracentrifugation for 3 hr at 38,000 rpm and 4°C in an SW41Ti rotor in a Beckman Coulter Optima XPN80 centrifuge. Fractions 6–13 of 20 total fractions were collected from the bottom of each tube, extracted using phenol:chloroform:isoamyl alcohol 25:24:1 (pH 5.2), ethanol precipitated, resuspended in 10 mM Tris (pH 8.0), and stored in −80°C.

## Sequencing library construction

Rec-Seq sequencing library construction was conducted according to a previously described protocol (*McGlincy and Ingolia, 2017*), with modifications, using identical barcoded linkers (NI-810 to NI-815), RT primer (NI-802) and PCR primers (NI-798, NI-799, NI-822 to NI-824). The RNA fragments purified from 48S PIC RPFs were dephosphorylated using T4 Polynucleotide Kinase (PNK) (NEB; M0201L) and ligated to pre-adenylated linkers (NI-810 to NI-815) containing 5 nt sample barcodes unique for each sample using truncated T4 RNA ligase 2 (K227Q) (NEB; M0351L). Ligated fragments were separated from free linkers on a 15% polyacrylamide TBE-Urea gel and then pooled and purified for reverse transcription using RT primer NI-802 and ProtoScript II Reverse Transcriptase (NEB; M0368S). The ~105 nt cDNAs were separated from free RT primers on a 15% TBE-Urea gel and circularized using CircLigaseII ssDNA Ligase (Biosearch Technologies; CL9021K). PCR was carried out using forward primer NI-798 and reverse primers (NI-799, NI-822-824) as already described (*McGlincy and Ingolia, 2017*).

RNA-Seq for input total mRNA was performed as previously described (*Ingolia et al., 2009*; *Ingolia et al., 2009*). Briefly, total mRNA was randomly fragmented at 70°C for 8 min in fragmentation reagent (Thermo Fisher; AM8740) and size-selected for 50–90 nt fragments for constructing a sequencing library using Universal miRNA Cloning Linker (NEB; S1315S) and the RNA-Seq library

**Table 1.** Oligonucleotides used for spike-in mRNA template amplification.

| Primer name | Sequence (5'–3') |
| --- | --- |
| T7-FLUC | AAGGAATTCATCTTAACTTTT<u>AATACGACTCACTATAGG</u>GCAAACAAACAAACCAAAACCACA***ATG***GAAGAGACGGCCAAAAACATA |
| T7-RLUC | AAGGAATTCATCTTAACTTTT<u>AATACGACTCACTATAGG</u>GCAAACAAACAAACCAAAACCACCACC***ATG***ACTTGCAAAGTTTATGAT |
| FLUC-R | GTCGACGAGGAATTCATTATCAGTGC |
| RLUC-R | GTCGACTTCTCCTTCTTCAGATTTGATC |

T7 promoter sequences are in bold and underlined; mAUGs are in bold and italicized.

construction procedures described above. Sequencing was done on an Illumina NovaSeq 6000 system at the NHLBI DNA Sequencing and Genomics Core at NIH (Bethesda, MD).

## Deep sequencing data processing and downstream analysis

The constant linker sequence (AGATCGGAAGAGCAC) in barcoded linkers was removed from Illumina NovaSeq reads using Cutadapt 4.0, and the mixed sample sequences were separated by the sample barcodes and aligned to the *S. cerevisiae* non-coding RNA genome using STAR 2.7.9a (*Dobin et al., 2013*) to remove non-coding RNA reads. The remaining RNA reads were then mapped to the reference genome (R64-1-1 S288C Sac cer3 Genome Assembly) and spike-in 'genome' using STAR 2.7.9a. Reads unaligned to the yeast genome were then mapped to the spike-in genome to obtain spike-in RPF counts. Similarly, reads unaligned to the spike-in genome were aligned to the yeast genome for genomic mRNA RPF counts. Size factors for each sample were calculated using the geometric means of the numbers of 25–34 nt reads mapping to both the mAUG and the many internal AUGs of the spike-in mRNAs, *FLUC* and *RLUC*. Consistent with previous reports (*Vandesompele et al., 2002*), it was essential to use the geometric mean rather than arithmetic mean for the spike-in normalizations, presumably because of the exponential nature of the PCR amplification step of library construction. All samples were normalized to the sample with the highest spike-in geometric mean among nine samples (three replicates each for 0 nm, 100 nm, and 500 nm Ded1). Size-factor-normalized wiggle tracks for each replicate or average of replicates were produced from the alignment file, one each for genes on the Watson or Crick strand. The sequences for the input total mRNA were processed similarly: after trimming NEB universal linker sequences and removing non-coding RNAs, the remaining reads were mapped to the yeast genome. Wiggle tracks were produced by assigning reads to the position of their 5′ ends. Two-dimensional metagene plots were produced by aligning all genomic mRNAs to their mAUGs and plotting the density of all RPFs based on footprint lengths and the positions of their 5′ ends map using a custom Python script (https://github.com/zhoufj/Metagene_plot copy archived at *Shin and Zhou, 2023*).

For counting RPFs on mRNAs, all 25–34 nt Rec-Seq reads were assigned to their predicted P-site mapped on the mRNA. The reads mapped between the –3 and +6 of the mAUG were counted as main RPFs (mRPFs), the reads counted between the mAUG and the stop codon of the CDS were counted as CDS RPFs (cdsRPFs), and the reads from the +9 position of the mAUG to the stop codon counted as internal RPFs (iRPFs). mRNA read counts were determined for all codons of the main CDS. DESeq2 (*Love et al., 2014*) was employed for differential expression analysis of changes in RPF, RE, or RRO values, and to impose cutoffs for minimum read numbers (as indicated in the figure legends) and remove outliers. Size factors calculated from spike-in RPFs were applied in DESeq2 analysis as customized size factors.

## Main AUG context scores

The AUG context adaptation index (context score) (*Miyasaka, 1999*) for all mAUGs with annotated 5′UTRs >5 nt were calculated previously (*Martin-Marcos et al., 2017*).

## PARS scores

The PARS scores, including total, Max30, Start30, Plus15, Plus30, and Plus45 PARS, were calculated as previously described (*Sen et al., 2015*) using the same dataset (*Kertesz et al., 2010*). One caveat with our analyses using PARS scores is that we did not subject our mRNA preparation to the final refolding protocol used by Kertesz et al. when they originally determined the scores. It is possible that we would have seen stronger correlations in the analyses using PARS scores had we followed their final renaturation protocol, although the fact that we observed significant correlations (e.g., *Figure 3E–H*) suggests the structures in the Kertesz et al. mRNAs were similar to those in our mRNAs.

## RE calculations

Reads of input RNA-Seq were counted from the main start codon to the stop codon and normalized to CDS length to calculate mRNA density. The REs for each mRNA in each condition were calculated as the ratio between the normalized mRPF value and mRNA density.

## Accession number

Sequencing data from this study have been submitted to the NCBI Gene Expression Omnibus (GEO; http://www.ncbi.nlm.nih.gov/geo/) under the accession number GSE244093. The ribosome profiling

data (*Sen et al., 2015*) reanalyzed in this study are under accession numbers GSM1621988 to GSM1621995 in the GSE66411 records.

## Acknowledgements

This work was supported by the Intramural Research Program of the *Eunice Kennedy Shriver* National Institute of Child Health and Human Development. We thank other members of our laboratories and those of Tom Dever and Nick Guydosh for many helpful suggestions. We thank Ryan K Dale from NICHD/NIH Bioinformatics and Scientific Programming Core and Byung-sik Shin for their help in sequencing data processing.

## Additional information

### Competing interests

Alan G Hinnebusch: Reviewing editor, eLife. The other authors declare that no competing interests exist.

### Funding

| Funder | Grant reference number | Author |
|---|---|---|
| Eunice Kennedy Shriver National Institute of Child Health and Human Development | 1ZIAHD008940 | Julie M Bocetti<br>Jon R Lorsch |
| Eunice Kennedy Shriver National Institute of Child Health and Human Development | 1ZIAHD001004 | Alan G Hinnebusch |

The funders had no role in study design, data collection and interpretation, or the decision to submit the work for publication.

### Author contributions

Fujun Zhou, Conceptualization, Data curation, Formal analysis, Validation, Investigation, Visualization, Methodology, Writing – review and editing; Julie M Bocetti, Formal analysis, Visualization, Writing – review and editing; Meizhen Hou, Resources, Methodology, Writing – review and editing; Daoming Qin, Conceptualization, Methodology, Writing – review and editing; Alan G Hinnebusch, Jon R Lorsch, Conceptualization, Supervision, Funding acquisition, Writing – original draft, Project administration, Writing – review and editing

### Author ORCIDs

Fujun Zhou ⓘ http://orcid.org/0000-0001-9772-7836
Jon R Lorsch ⓘ https://orcid.org/0000-0002-4521-4999

Reviewer #1 (Public Review): https://doi.org/10.7554/eLife.93255.3.sa1
Reviewer #2 (Public Review): https://doi.org/10.7554/eLife.93255.3.sa2
Author response https://doi.org/10.7554/eLife.93255.3.sa3

## Additional files

### Supplementary files

• MDAR checklist

### Data availability

Sequencing data from this study have been submitted to the NCBI Gene Expression Omnibus (GEO; http://www.ncbi.nlm.nih.gov/geo/) under the accession number GSE244093. Custom scripts used in

data analysis are available at https://github.com/zhoufj/Metagene_plot_1.1 (copy archived at *Zhou, 2024*).

The following dataset was generated:

| Author(s) | Year | Dataset title | Dataset URL | Database and Identifier |
| --- | --- | --- | --- | --- |
| Zhou F, Bocetti JM, Hou M, Qin D, Hinnebusch AG, Lorsch JR | 2023 | Transcriptome-wide analysis of the function of Ded1 in translation preinitiation complex assembly in a reconstituted in vitro system | https://www.ncbi.nlm.nih.gov/geo/query/acc.cgi?acc=GSE244093 | NCBI Gene Expression Omnibus, GSE244093 |

The following previously published dataset was used:

| Author(s) | Year | Dataset title | Dataset URL | Database and Identifier |
| --- | --- | --- | --- | --- |
| Sen ND, Zhou F, Ingolia NT, Hinnebusch AG | 2015 | Genome-wide analysis of translational efficiency reveals distinct but overlapping functions of yeast DEAD-box RNA helicases Ded1 and eIF4A | https://www.ncbi.nlm.nih.gov/geo/query/acc.cgi?acc=GSE66411 | NCBI Gene Expression Omnibus, GSE66411 |

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
