## [Editor Report · eLife assessment]

This is an **important** article as it is the first to use a reconstituted translation system to study competition among mRNAs for the initiation machinery. Understanding the principles of the biochemistry of mRNA competition for initiation factors cannot be achieved without such a system. The authors provide **compelling** evidence that Ded1 is required for efficient initiation of highly structured mRNAs. The findings are significant and validate the in vitro reconstituted system by recapitulating the effects of in vivo perturbations of translation initiation by Ded1 mutants.

---

## [Referee Report · Reviewer #1 (Public Review)]

The authors have developed and optimized a footprinting assay to monitor the recruitment of mRNAs to a reconstituted translation initiation system. This assay is named Recruitment-Sequencing (Rec-Seq) and enables the analysis of many purified mRNAs in the reconstituted system.

This system possesses the ability to determine how competition occurs between mRNAs for the initiation machinery. This is the first approach using a reconstituted system that enables this important feature, and this is an important advance for the field.

Using purified mRNAs in a fully reconstituted system together with the ability to monitor start site selection is an important advance. The method enables one to observe for the first time how competition between mRNAs is altered in response to the absence or presence of different initiation components or accessory proteins.

Start site fidelity in purified reconstituted systems can be altered in different buffer conditions and by the concentration of various initiation factors involved in start site fidelity. Future experiments will reveal how these variables can regulate start site selection in this powerful system.

Comments on revised version:

The authors have addressed all of my original comments. This is an impressive manuscript.

---

## [Referee Report · Reviewer #2 (Public Review)]

Summary:

Zhou et al report development of a new method, Rec-Seq, that allows rigorous quantitation of the efficiency of 48S ribosomal pre-initiation complex (PIC) formation on messenger RNAs at transcriptome scale in vitro. With a next-generation deep-sequencing approach, Rec-Seq allows precisely targeted dissection of the roles of translation initiation factors in PIC assembly. This level of molecular precision is important to understanding mechanisms of translational control, making Rec-Seq a significant methodological advance. The authors leverage Rec-Seq to investigate the relative roles of two key helicase enzymes, Ded1p and eIF4A. While past work has pointed to differing roles for Ded1p and eIF4A helicase activity in PIC assembly, unambiguous interpretation of prior in-vivo data has been hindered by technical requirements for performing the experiments in cells. Rec-Seq circumvents these challenges, providing robust mechanistic insights. The authors find that Ded1p stimulates PIC formation selectively on mRNAs with long, structured leaders in the Rec-Seq system, while eIF4A provides much more general stimulation across mRNAs. The findings substantiate the past in-vivo results, along with adding new insights. They contrast with evidence that Ded1p promotes translation by suppressing inhibitory upstream initiation through structural remodeling, or through formation of intracellular, phase-separated granules. The conclusions of the study are well-supported by the data, and are likely to be of broad interest.

Strengths:

The quantitative nature of Rec-Seq, which uses an internal standard to measure absolute recruitment efficiencies, is an important strength.

The methodology decisively overcomes past experimental limitations, allowing the authors to make clear conclusions with regard to the relative roles of Ded1p and eIF4A in PIC formation. An important and useful addition to the toolbox for studying translation and translational control mechanisms, Rec-Seq substantially expands the throughput and scope of mechanistic analyses for translation initiation.

One significant finding to emerge is that the in-vitro reconstituted system used here recapitulates effects of in-vivo perturbations of translation initiation. Despite the lack of a cellular environment and its components, PIC formation appears to operate much as it does in the cell. Importantly, this highlights an inherent "modularity" to the system that is especially of interest in the context of how regulatory machinery beyond the PIC may control translation.

Weaknesses:

The study finds that Ded1p stimulates accumulation of PICs at internal AUG codons, i.e., within mRNA coding sequences, at an incidence of up to ~50% - thus, bypassing "canonical" translation start sites. Understanding the physiological significance of this activity will require further study. The authors address this in the text.

A limitation of the methodology is that, as an endpoint assay, Rec-Seq does not readily decouple effects of Ded1p on PIC-mRNA loading from those on the subsequent scanning step where the PIC locates the start codon. Considering that Ded1p activity may influence each of these initiation steps through distinct mechanisms - i.e., binding to the mRNA cap-recognition factor eIF4F, or direct mRNA interaction outside eIF4F - additional studies will be needed to gain deeper mechanistic insights. The authors discuss this in the text.

Comments on revised version:

In revising their manuscript, the authors have responded very thoughtfully and insightfully to the initial review. The final manuscript is an important contribution to the field, and I am sure it will be of broad interest.

---

## [Author Response]

The following is the authors’ response to the original reviews.

**Reviewer #1**
Weaknesses:Start site fidelity in purified reconstituted systems can be drama5cally altered in different buffer conditions. Interpretation of the observed changes to start site selection in mRNAs in the absence or presence of Ded1 using only the one buffer condition used is therefore limited.

This is an excellent point and is something we could explore in future studies using the Rec-Seq system. We have added this caveat to the Discussion on lines 797-809. We have previously studied the fidelity of start codon recognition in the reconstituted system (Kolitz et al., [2009] RNA, 15:138-152) and found that under our standard buffer conditions the codon specificity generally reflects what we observed in vivo using a dual-luciferase reporter assay, with the most stable 48S complexes forming on AUG codons, followed by first position mismatches (GUG, UUG, CUG), with second and third position mismatches leading to significantly less stable complexes. However, as the reviewer notes, there are some deviations: ACG and AUA are poor codons in the in vitro system under the buffer conditions used but allowed relatively strong expression in our in vivo reporter assay. It should also be noted that the hierarchy of nearcognate start codon usage in vivo in yeast differs according to the study and the reporter used, making it difficult to establish a “ground truth” for start codon fidelity.

I have some specific comments to strengthen the manuscript and address some minor issues.It is not clear to me whether the authors refold the purified mRNA aEer phenol/chloroform extraction? Have the authors observed different results if the mRNA is refolded or not? This is appropriate since the authors compare their Rec-Seq data to PARS scores that were generated from refolded mRNAs. One assumes that the total mRNA used is refolded in the same way as the PARS score study, but this is not clearly stated. The authors should make this point clear in the text and methods.

This is an excellent point. We did not use the final refolding protocol that Kertesz et al. used when they developed their PARS scores and now clarify this in the Methods section (lines 962967). It is possible that we would have seen stronger correlations in the analyses using PARS scores had we followed the renaturation protocol, although the fact that we observed significant correlations (e.g., Fig. 3E-H) suggests the structures in the Kertesz et al. mRNAs were similar to those in our mRNAs.

It is not clear how the authors determine the concentration of total mRNA that is used in the assay - reported as 60 nM? Are the authors assuming a molecular weight of an average mRNA to determine the concentration? The authors should provide more detail for how they quantify their mRNA concentration and its stoichiometry compared to 43S PICs.

We thank the reviewer for pointing out this oversight and have now included this information on lines 849-855 of the Methods section.

Comments regarding start site fidelity in the reconstituted system:The authors use in vitro transcribed tRNAi-Met. Since tRNA modifications may play a role in start site fidelity, the authors should perhaps mention that this will need to be investigated in a future study in the discussion.

This is a good point and we now note it as a caveat in the Discussion on lines 806-809.

The authors state that Ded1 promotes leaky scanning regardless of the mAUG start site context (page 24; lines 533-534). The authors then state on page 25 that the level of iAUG initiation relative to mAUG initiation does depend on the mAUG context (lines 545-546). This seems contradictory unless I am not understanding this correctly? It would certainly be surprising that mAUG context didn't regulate leaky scanning in the reconstituted system given the fact that initiation codon context regulates selection in cells (when Ded1 is present).

These statements are correct as wrihen. As shown in Figure 5O, the frequency of leaky scanning (as measured by relative ribosome occupancy of the internal region of the ORF, not including the main start codon, to the whole ORF, including the main start codon; RRO) decreases as the context score around the start codon gets stronger (green and purple lines). The RRO is increased to the same extent when 500 nM Ded1 is added, regardless of the strength of the start codon context, indicating that Ded1 enhances leaky scanning equally (compare slopes of the green line without Ded1 to the purple line with Ded1). Because of this, the effect of Ded1on RRO (DRR0) is constant across context score bins (orange line). There is no discrepancy between our two conclusions that leaky scanning of the mAUG increases as context score decreases and that Ded1 increases leaky scanning equally for good and bad mAUG contexts, indicating that Ded1 does not inspect the mAUG context and simply decreases the dwell time equally at all contexts.

Further to the start site context question. It is possible that the fidelity of the reconstituted system (i.e. buffer conditions) is not fully reflecting in vivo-like start site selection. A rigorous characterization of commercially available reticulocyte lysate systems identified buffer conditions that provided similar start site fidelity to that observed in live cells (Kozak. Nucleic Acids Res. 1990 May 11;18(9):2828). While I feel that it is beyond the context of the current work to undertake a similar rigorous buffer characterization, one must be careful about interpreting the results about leaky scanning and upstream initiation sites in the current work. Perhaps one would observe similar results to Guenther et al. if the fidelity (buffer conditions) of the reconstituted system were different? I appreciate that the authors state that their results only apply to their reconstituted system and do not necessarily suggest that previous data are incorrect, but with only one buffer condition being tested in the current study it may be appropriate to further soEen the interpretation of the current results when compared to published data in live cells.

This point is well-taken. As noted above, we have added a caveat about possible effects of buffer conditions on start codon fidelity to the Discussion (lines 797-809). In terms of the possibility that upstream initiation is more frequent in vivo than we observe in the in vitro RecSeq system, we previously studied 5′UTR translation in vivo using ribosome profiling (Kulkarni et al. [2019] BMC Biol., 17:101). The ratio of RPFs in 5′UTRs to coding sequences in this study was0.0027, very similar to the value measured in the in vitro Rec-Seq system in the presence of Ded1 (0.0016-0.0017). Thus, it does not seem that the frequency of upstream initiation is dramatically higher in vivo than in our in vitro system. We have now made note of this point in the Results (lines 594-598). Guenther et al. employed a ribosome profiling protocol in which they added cycloheximide to their cells prior to lysis, which has been shown to create significant artifacts, particularly in 5′UTR translation (e.g., Gerashchenko and Gladyshev [2014] Nucleic Acids Res., 42:e134). Nevertheless, as suggested by the reviewer, we have modified the text in the Results and Discussion to somen the interpretation somewhat (lines 582-583; 616-618; 761763).

**Reviewer #2**
Weaknesses:Several findings in this report are quite surprising and may require additional work to fully interpret. Primary among these is the finding that Ded1p stimulates accumulation of PICs at internal site in mRNA coding sequences at an incidence of up to ~50%. The physiological relevance of this is unclear.

We agree with the reviewer that understanding the physiological significance, if any, of the apparent leaky scanning of main AUG start codons induced by Ded1 is an unanswered question that will require additional studies. It is possible that rapid 60S subunit joining and formation of the 80S initiation complex amer start codon recognition on most mRNAs reduces the leaky scanning effect in vivo. We now bring up this possibility in the Discussion section (lines 804809). However, as noted in lines 568-580, mRNAs that display significantly decreased mRPFs at 500 nM Ded1 in the Rec-Seq system also tend to have TEs that are increased in the ded1-cs- mutant relative to WT yeast in in vivo ribosome profiling experiments, suggesting that Ded1 activity also diminishes initiation on mAUG codons in these mRNAs in vivo.

A limitation of the methodology is that, as an endpoint assay, Rec-Seq does not readily decouple effects of Ded1p on PIC-mRNA loading from those on the subsequent scanning step where the PIC locates the start codon. Considering that Ded1p activity may influence each of these initiation steps through distinct mechanisms - i.e., binding to the mRNA cap-recognition factor eIF4F, or direct mRNA interaction outside eIF4F - additional studies may be needed to gain deeper mechanistic insights.

We agree that this is a limitation of the Rec-Seq assay and now mention this point in the Discussion section (lines 810-817). It is possible that future work using cross-linking agents to stabilize 43S complexes bound near the cap and scanning the 5′UTR, similar to the methodology used in 40S ribosome profiling, could enable us or others to disentangle these steps from one another.

As the authors note, the achievable Ded1p concentrations in Rec-Seq may mask potential effects of Ded1p-based granule formation on translation initiation. Additional factors present in the cell could potentially also promote this mechanism. Consequently, the results do not fully rule out granule formation as a potential parallel Ded1p-mediated translation-inhibitory mechanism in cells.

We agree. As stated in the Discussion section (lines 735-741): “It is possible that at higher concentrations of Ded1 than were achievable in these in vitro experiments or in the presence of additional factors that modify Ded1’s ATPase or RNA binding activities the factor could directly inhibit a subset of mRNAs, by acting as an mRNA clamp that impedes scanning by the PIC, or by sequestering the mRNAs in insoluble condensates. It might be interesting in the future to test candidate factors in Rec-Seq to determine if they switch Ded1 from being a stimulatory helicase to an inhibitory mRNA clamp that removes transcripts from the soluble phase.”

It is certainly clear why the 15-minute timepoint was chosen for these assays. However, I wondered whether data from an earlier timepoint would provide useful information. The description on line 210 of the compiled PDF suggests data from different timepoints may be available; if it is, in my view it could be a useful addition. More generally, including language about the single-turnover nature of these reactions may be helpful for the benefit of a broad audience.

In preliminary experiments, we have used the Rec-Seq system to measure the kinetics of 48S PIC formation transcriptome-wide. As you probably can imagine, this is a challenging experiment and requires additional work before we would feel comfortable publishing it. We very much agree with the reviewer that resolving the kinetics of these events will provide important additional information. As suggested, we have added caveats about the endpoint and single-turnover nature of the assay to the Discussion (lines 821-828).

I wondered whether it might be useful to present additional information on the mRNAs not found in the assay. For example, are these the least abundant mRNAs, which may not have had time to recruit the 43S PIC?

75% of mRNAs (2719 of 3640) not observed in the Rec-Seq analysis had densities below the median (2.3 reads per nucleotide). We now mention this in the Methods section (lines 855856).

The Rec-Seq recruitment reactions were carried out at 22C° . Considering that remodeling of RNA structure by helicase enzymes is a focal point of the study, linking the results to the recruitment landscape at a closer-to-physiological temperature may bolster the conclusions.

In the future, it would be interesting to test the effects of temperature on 48S PIC formation using the Rec-Seq system. As the reviewer suggests, the interplay between temperature and mRNA structure could reveal interesting phenomenon. It is worth noting, however, that there is no clear “physiological” temperature for *S. cerevisiae*. For consistency and convenience, lab yeast is usually grown at 30 °C, but in the wild yeast live at a wide range of temperatures, which generally change throughout the day. From this standpoint, 22 °C seems reasonably physiological.

Results from Rec-seq experiments conducted at 15° C might be more directly comparable to in vivo Ribo-seq data with the ded1-cs mutant. However, already ~90% of the Ded1hyperdependent mRNAs identified by Ribo-seq analysis of that mutant were identified here as Ded1-stimulated mRNAs in Rec-Seq experiments at 22°C. The Ribo-seq experiments conducted by Guenther et al. were conducted on the ded1-ts mutant at 37°C; thus, any structures that confer Ded1-dependent leaky-scanning through uORFs detected in that study should have been stable in our Rec-Seq experiments.

The introduction provides an important, detailed exposition of the state of the field with respect to Ded1p activity. Nevertheless, in my view, it is quite lengthy and could be streamlined for clarity. As just one example, the proposed function of Ded1p in the nucleus seems like a detail that could be dispensed with for the present work.

We have ahempted to shorten the Introduction, as suggested. However, we did not remove the short section describing Ded1’s possible roles in the nucleus and ribosome biogenesis because we felt it was important to emphasize that one of the strengths of the Rec-Seq system is that it allows us to isolate the early steps of translation initiation from later steps and from other cellular processes. In addition, at the suggestion of Reviewer #3, we added a brief explanation of Ded1’s possible role in the subunit joining step of translation.

**Reviewer #3**
Weaknesses:The slow nature of the biochemical experiments could bias results.

We agree that the 15-minute time point used could mask effects that are manifested at a purely kinetic level. It should be noted that we have measured the observed rate constants for 48S formation on a variety of mRNAs in the in vitro reconstituted system in the presence of saturating Ded1 (Gupta et al. [2018] eLife, https://elifesciences.org/articles/38892 ) and found that they are generally in the range of estimates of rate constants for translation initiation in vivo in yeast (~1-10 min-1; e.g., Siwiak and Zielenkiewicz [2010], PLOS Comput. Biol., 6: e100865). In preliminary experiments, we have used the Rec-Seq system to measure the kinetics of 48S PIC formation transcriptome-wide in the absence of Ded1 and find that the mean rate constant observed (~2 min-1) is also within the range of estimates of the rate of translation initiation in vivo in yeast. We hope to publish this analysis in a future manuscript.

It has been suggested that Ded1 and its human homolog DDX3X could play a role in subunit joining postscanning (Wang et al. 2022, Cell and Geissler et al. 2012 Nucleic Acids Res). Could the authors potentially investigate this by adding GTP, eIF5B and 60S subunits into the reaction mixture and isolating 80S complexes?

This is a very interesting suggestion. One of our plans with the Rec-Seq system is to see if we can also observe 80S formation with it and distinguish 80S from 48S complexes. Although we haven’t yet tried this and there might be technical obstacles to doing it, if it works we would like to examine the potential effects of Ded1, as suggested. We now mention this possibility in the Discussion section (lines 709-716 and 810-817).

An incubation time of 15 minutes is quite long on the timescale of translation initiation. Presumably, the competion for 40S among mRNAs is partially kinetically controlled so it would be interesting if the authors could do a time series on the incubation time. Does Ded1 increase initiation on more structured UTRs even at shorter incubations or are those only observed with longer incubations?

We agree. See the response to the question about kinetics above.

Does GDPNP lead to off-pathway events? What happens when GTP is used in the TC? Presumably in the absence of eIF5B the 48S PIC should remain stalled at the start codon.

In previous experiments in the reconstituted system, we showed that using GTP instead of GDPNP resulted in 48S complexes that were less stable than those stalled prior to GTP hydrolysis (e.g., Algire et al. [2002] RNA 8:382-397). This is presumably because eIF2•GDP and eIF5 release from the complex and the Met-tRNAi can dissociate in the absence of subunit joining. Although we haven’t tried it in the Rec-Seq system, we suspect that the resulting PICs would fall apart during sucrose gradient sedimentation.

The authors use assembly of a 48S PIC at the start codon as evidence of scanning but could use more evidence to back this claim up. Does removing the cap structure on the two luciferase mRNA controls disrupt initiation using this approach? That would be direct evidence of 5' end 40S loading and scanning to the start codon.

In previous work using the reconstituted system, we studied the effect of the 5′-cap on 48S PIC formation (Mitchell et al. [2010] Mol. Cell 39:950-962; Yourik et al. [2017] eLife https://elifesciences.org/articles/31476 ). We found that stable 48S PIC formation is strongly dependent on the presence of the 5′-cap. In addition, the cap prevents off-pathway events and enforces a requirement for the full set of initiation factors to achieve efficient 48S PIC formation. As the reviewer indicates, the cap-dependence of the system supports the conclusion that 5′end loading and scanning take place. We have now added this information and the relevant citations to the Introduction (lines 147-153). We thank the reviewer for pointing out this oversight. It should also be noted that the cases of mRNAs in which 5′UTR translation is increased by addition of Ded1 support the conclusion that the factor promotes ahachment of the PIC to the 5′ ends of mRNAs and subsequent 5′ to 3’ scanning, as noted in lines 608-618.

The authors state that "The correla5on between CDS length and RE could be indirect because CDS length also correlates with 5'UTR length". Could the authors bin the transcripts into different 5' UTR length ranges and then probe for CDS length differences on RE for each 5' UTR length bin? This could be useful to truly parse the mechanism by which CDS length is influencing RE.

This was an excellent suggestion. We now include this analysis in a new supplementary figure, Figure 3S-2. Corresponding text was added in lines 380-387:

“Importantly, correlations between Ded1 stimulation and 5′ UTR lengths are evident for all three groups of mRNAs containing distinct ranges of CDS lengths (Fig. 3-S2A-C). In contrast, a marked correlation between Ded1 stimulation and CDS length was detected only for the group of mRNAs with longest 5′UTRs (Fig. 3-S2D-F), and only the latter group showed a clear correlation between 5′UTR length and CDS length (Fig. 3-S2G-I). Thus, the correlation between Ded1 stimulation and CDS length appears to be indirect, driven by the tendency for the mRNAs with the longest 5′UTRs to also have correspondingly longer CDSs.”

We thank the reviewer for this very useful idea.

In Figure 3I, why does RE dip for the middle bins of CDS length in both 100 nM and 500 nM conditions, and then rise back up for the later bins? In other words, why do the shortest and longest CDS have the best RE in the presence of ded1?

We do not know the reason for this dip and now say this in the Results on lines 377-378.

The discussion section would be well served to discuss proposed roles of Ded1 post-scanning and how those fit, if at all, with the data presented throughout the manuscript.

We have now added this to the Discussion (lines 709-716 and 810-817). We thank the reviewer for pointing out this oversight.

Minor comments:Define bins on figures rather than using bin number for axis labels. For example, Figure 3A-D x-axis labels indicate the length range of each bin.

Thank you for the suggestion. We have made this change.

Figure 3I: the data seem to indicate that shortest CDSs have a ded1 dependency similar to the longest CDSs. This result seems inconsistent with the given relationship between UTR length, structure, CDS length. Please clarify.

See answer to this ques>on above.

Replace qualitative statements, such as "substantially smaller reductions" with percent change, numbers, etc.

We have tried to replace qualitative statements with quantitative ones, where possible.